# Acute perturbation of *Pet1*-neuron activity in neonatal mice impairs cardiorespiratory homeostatic recovery

Ryan T Dosumu-Johnson[1], Andrea E Cocoran[2], YoonJeung Chang[1], Eugene Nattie[2], Susan M Dymecki[1]*

[1]Department of Genetics, Harvard Medical School, Boston, United States; [2]Department of Molecular & Systems Biology, The Geisel School of Medicine at Dartmouth, Hanover, United States

**Abstract** Cardiorespiratory recovery from apneas requires dynamic responses of brainstem circuitry. One implicated component is the raphe system of *Pet1*-expressing (largely serotonergic) neurons, however their precise requirement neonatally for homeostasis is unclear, yet central toward understanding newborn cardiorespiratory control and dysfunction. Here we show that acute in vivo perturbation of *Pet1*-neuron activity, via triggering cell-autonomously the synthetic inhibitory receptor hM4D$_i$, resulted in altered baseline cardiorespiratory properties and diminished apnea survival. Respiratory more than heart rate recovery was impaired, uncoupling their normal linear relationship. Disordered gasp recovery from the initial apnea distinguished mice that would go on to die during subsequent apneas. Further, the risk likelihood of apnea-related mortality associated with suppression of *Pet1* neurons was higher for animals with baseline elevated ventilatory equivalents for oxygen. These findings establish that *Pet1* neurons play an active role in neonatal cardiorespiratory homeostasis and provide mechanistic plausibility for the serotonergic abnormalities associated with SIDS.

DOI: https://doi.org/10.7554/eLife.37857.001

*For correspondence: dymecki@genetics.med.harvard.edu

**Competing interests:** The authors declare that no competing interests exist.

## Introduction

Tissue oxygen levels are maintained within a narrow, life-sustaining range through the coordinated actions of oxygen flux through breathing, red blood cell loading-unloading, and circulation via heart pumping. This vital cardiorespiratory homeostasis is subserved by the interplay of neural circuits that must reliably function for the duration of an organism's *ex utero* life. During early neonatal life, mammals are especially vulnerable to homeostatic impairments given their relatively lower oxygen reserve due to smaller lung volume and faster decline in blood oxygen levels during hypoxic conditions owing to the steep desaturation kinetics of the still presiding fetal oxyhemoglobin (*Fewell, 2005*). Paradoxically, young mammals including human infants undergo more frequent interruptions in breathing, called apneas, that are typically coupled with heart rate slowing, bradycardias (*Daily et al., 1969*; *Fewell et al., 2005*; *Kelly et al., 1985*; *Southall et al., 1980*). At the same time, neonatal mammals have an especially robust, protective homeostatic response – referred to as autoresuscitation – which utilizes gasping after an apnea to increase blood oxygen levels to facilitate restoration of heart rate and eupneic breathing (*Gershan et al., 1992*; *Guntheroth and Kawabori, 1975*; *Jacobi et al., 1991*; *Saiki et al., 2001*). Here we present progress in delineating aspects of the underlying neurobiology, querying the neonatal importance for *Pet1*-lineage neurons, largely serotonergic (5-hydroxytryptamine-, 5-HT-producing), in the apnea recovery response.

Serotonergic transmission has been implicated in the autoresuscitation response in rodents (*Barrett et al., 2016*; *Cummings et al., 2011a*; *Erickson and Sposato, 2009*; *Sridhar et al., 2003*)

**eLife digest** Our survival depends on our heart and lungs working together to supply our cells with oxygen and remove carbon dioxide waste. The brain coordinates this process by controlling the activity of the heart and lungs. Yet sometimes a person may experience an event called an apnea and briefly stop breathing. If this happens, oxygen levels in the body fall while carbon dioxide levels rise. This in turn triggers a recovery process called autoresuscitation, which includes a series of large breaths or gasps, and each gasp is accompanied by increased heart rate due to specialized parts of the nervous system. This response usually restores normal breathing.

Failure of autoresuscitation may underlie many cases of sudden infant death syndrome, or SIDS (also known as "cot death" or "crib death"). SIDS is the leading cause of death in young infants in the western world, and many infants who die from SIDS show abnormalities in the brain cells that produce a chemical called serotonin. Evidence suggests that serotonin helps control breathing. This raised the question: does the autoresuscitation recovery response rely on serotonin-producing neurons?

To find out, Dosumu-Johnson et al. used one-week-old mouse pups that had been genetically engineered to respond to an injected drug by rapidly inhibiting their serotonin neurons. These animals are about the same age in mouse terms as infants at greatest risk for SIDS (~2-4 months of age). Inhibiting serotonin neurons made it harder for the mouse pups to recover from artificially induced apneas. Although their heart rate showed largely normal recovery – at least at first – their breathing did not. They took fewer gasps, and were more likely to die following such episodes.

These findings shed new light on how young animals control their breathing and heart rate when mounting an autoresuscitation recovery from an apnea. The observed uncoupling of breathing and heart rate recovery responses suggests that different brain cells and circuits control the two. The results also suggest that abnormalities in the activity of serotonin neurons may make infants more susceptible to SIDS. As well as offering a possible explanation to families who have lost a child to SIDS, these findings could be used to develop screening tools to identify other infants at risk. They also point to potential cellular targets for drugs that could ultimately help prevent further cases.
DOI: https://doi.org/10.7554/eLife.37857.002

and in cardiorespiratory modulation in humans and rodents, including in the sudden infant death syndrome (SIDS) (*Duncan et al., 2010*; *Feldman et al., 2003*; *Hodges and Richerson, 2010*; *Kinney and Thach, 2009*; *Peña and Ramirez, 2002*; *Ptak et al., 2009*; *Ray et al., 2011*). In mice, chronically disabling vesicular neurotransmission from 5-HT neurons ('silencing' them) from mid-embryogenesis onward results in pups with diminished capacity to recover from induced asphyxic apnea (*Barrett et al., 2016*). Impairment was observed in pups across postnatal (P) days 5–8 but no longer by P12, suggestive of a neonatal period of heightened vulnerability to neurological dysfunction and cardiorespiratory stressors. Rodent pups with 80–90% reduction in medullary 5-HT content, resulting from perturbation chemically [5,7-dihydroxytryptamine treatment (*Cummings et al., 2011b*)] or genetically [germ line *Pet-1* gene deletion (*Cummings et al., 2011a*; *Erickson and Sposato, 2009*; *Erickson et al., 2007*) or *tryptophan hydroxylase* 2 (*Tph2*) deletion (*Chen et al., 2013*)] also showed impaired recovery from apneic challenges. These rodent data build mechanistic plausibility for the SIDS-associated findings of postmortem brainstem 5-HT neuron abnormalities (*Duncan et al., 2010*; *Paterson et al., 2006*), cardiorespiratory tracings showing prolonged and more frequent apneic and bradycardic events associated with progression leading to death (*Meny et al., 1994*; *Poets et al., 1999*; *Sridhar et al., 2003*), and the epidemiological determination of a postneonatal critical period of heightened SIDS risk (2–4 months of age)(*American Academy of Pediatrics Task Force on Sudden Infant Death Syndrome, 2005*). Yet evidence for an acute, real-time role for postneonatal serotonergic neurons in modulation of the autoresuscitation response remains lacking. Studies have largely involved chronic or extended 5-HT system manipulations spanning embryonic (*Barrett et al., 2016*; *Chen et al., 2013*; *Cummings et al., 2011a*; *Erickson and Sposato, 2009*) and/or postneonatal development (*Yang and Cummings, 2013*) in which secondary, compensatory network changes can occur in addition to the primary, engineered serotonergic neuronal abnormality. Here we report progress in this area through studies in which we acutely induced

*Pet1*-neuron perturbation in vivo at P8 and measured cardiorespiratory outcome and recovery across a chain of asphyxic-induced apneas.

We used an inducible (clozapine-N-oxide (CNO)-triggered) neuronal inhibition strategy (*Ray et al., 2011*) involving the cognate, synthetic inhibitory G protein-coupled receptor hM4D$_i$ (also referred to as D$_i$) (*Armbruster et al., 2007*) to disrupt at P8 the activity of a raphe neuron population defined by expression of a *Pet1* BAC transgene. *Pet1* gene expression serves largely as a serotonergic marker (*Fyodorov et al., 1998*), with the *Pet1* BAC as a driver offering genetic access to 5-HT neurons (5-HT$^+$, *Pet1$^+$*, tryptophan hydroxylase 2 (Tph2$^+$) cells), plus a small subset of raphe neurons that are negative for 5-HT while nonetheless positive for *Pet1* expression (5-HT$^-$, Tph2$^{low}$, *Pet1$^+$*-cells) (*Barrett et al., 2016*; *Okaty et al., 2015*; *Pelosi et al., 2014*; *Sos et al., 2017*). Our results suggest that *Pet1*-neuron activity is required neonatally for maintaining baseline heart rate and ventilation and for normal survival rates in response to apneas. Indeed, CNO-Di-mediated perturbation of *Pet1$^+$* neurons at P8 renders pups significantly more likely to die after an apnea when compared to CNO-treated sibling controls. Further, we found that this acute manipulation of *Pet1$^+$* neurons primarily affected the respiratory components of apnea recovery while sparing much of the cardiac response – a cardiorespiratory uncoupling not predicted by earlier chronic perturbation studies (*Barrett et al., 2016*; *Cummings et al., 2011a*; *Cummings et al., 2011b*) and which runs counter to the linear relationship between breathing and heart rate recovery present in control pups. Additionally, we found that a disordered gasp response to the first apnea characterized pups that succumbed to a subsequent apnea in the assay. As well, *post hoc* analyses identified specific respiratory features associating with autoresuscitation failure. These findings support a model in which *Pet1*-neuron activity is required neonatally for robust apnea recovery and may, by extension, inform strategies for pediatric autoresuscitation and SIDS prevention.

## Results

### P8 mouse neonates show altered baseline cardiorespiratory parameters in response to Di-mediated perturbation of *Pet1* neurons

To study *Pet1$^+$* neurons neonatally, we pursued a chemogenetic Di strategy allowing for noninvasive, inducible, targeted neuronal perturbation suitable to the small size of P8 mouse pups and the physical and temporal constraints of our plethysmographic apnea-induction-recovery (autoresuscitation) assay. We applied the Flpe recombinase-encoding BAC transgenic driver *Pet1-Flpe* (*Jensen et al., 2008*) acting on the engineered Flp-responsive *ROSA26 (R26)* allele designated *Gt(ROSA)26Sor$^{tm}$*$^{(CAG-FSF-CHRM4*(Di))Dym}$ (denoted for ease in short-hand as RC-FDi) (*Brust et al., 2014*; *Ray et al., 2011*) to drive in *Pet1* neurons expression of Di (*Armbruster et al., 2007*), the Gi/o protein-coupled receptor with engineered selectivity for the injectable synthetic ligand CNO. Established previously, CNO-triggering of Di signaling in *Pet1*-lineage neurons using this transgenic approach results in hyperpolarization and diminished excitability in vivo and in vitro (*Brust et al., 2014*; *Ray et al., 2011*; *Teissier et al., 2015*). Additionally, CNO-triggered Di signaling has been shown to inhibit synaptic transmission (*Stachniak et al., 2014*). The *Pet1-Flpe* transgene drives Flpe expression from mid-embryogenesis onward in a majority of *Pet1*-expressing postmitotic neurons (*Jensen et al., 2008*) and reliably mediates recombination of RC-FDi (*Ray et al., 2011*) and other *R26* engineered alleles (*Brust et al., 2014*; *Jensen et al., 2008*). Thus double transgenic *Pet1-Flpe*, RC-FDi pups allow for acute, inducible perturbation of *Pet1* neurons for neonatal, whole-animal functional study before and following intraperitoneal (i.p.) CNO administration. Immunohistochemically-stained sections of the raphe from P8 double transgenic *Pet1-Flpe,* RC-FDi mouse pups, referred to as *Pet1-Di* pups, confirmed protein expression of the HA-tagged Di receptor in serotonergic (Tph2$^+$) neurons (*Figure 1A–C'''*), consistent with previously published *Pet1-Flpe* driver specificity in neonatal pups (*Barrett et al., 2016*). The raphe location, proportion, and intensity of HA-Di immunodetection signal was qualitatively similar across *Pet1-Di* pups from independent litters (*Figure 1—figure supplement 1*). In sibling control single transgenic RC-FDi pups, referred to as control-Di pups (harboring the unrecombined RC-FDi allele and thus negative for *Di* transcription), no HA-Di immunosignal was detected (*Figure 1D–F*).

To assay cardiorespiratory function in P8 mouse pups under conditions of room air (RA) (*Figure 1G*, open rectangle *a*) and then apnea-inducing, asphyxia conditions (*Figure 1G*, filled

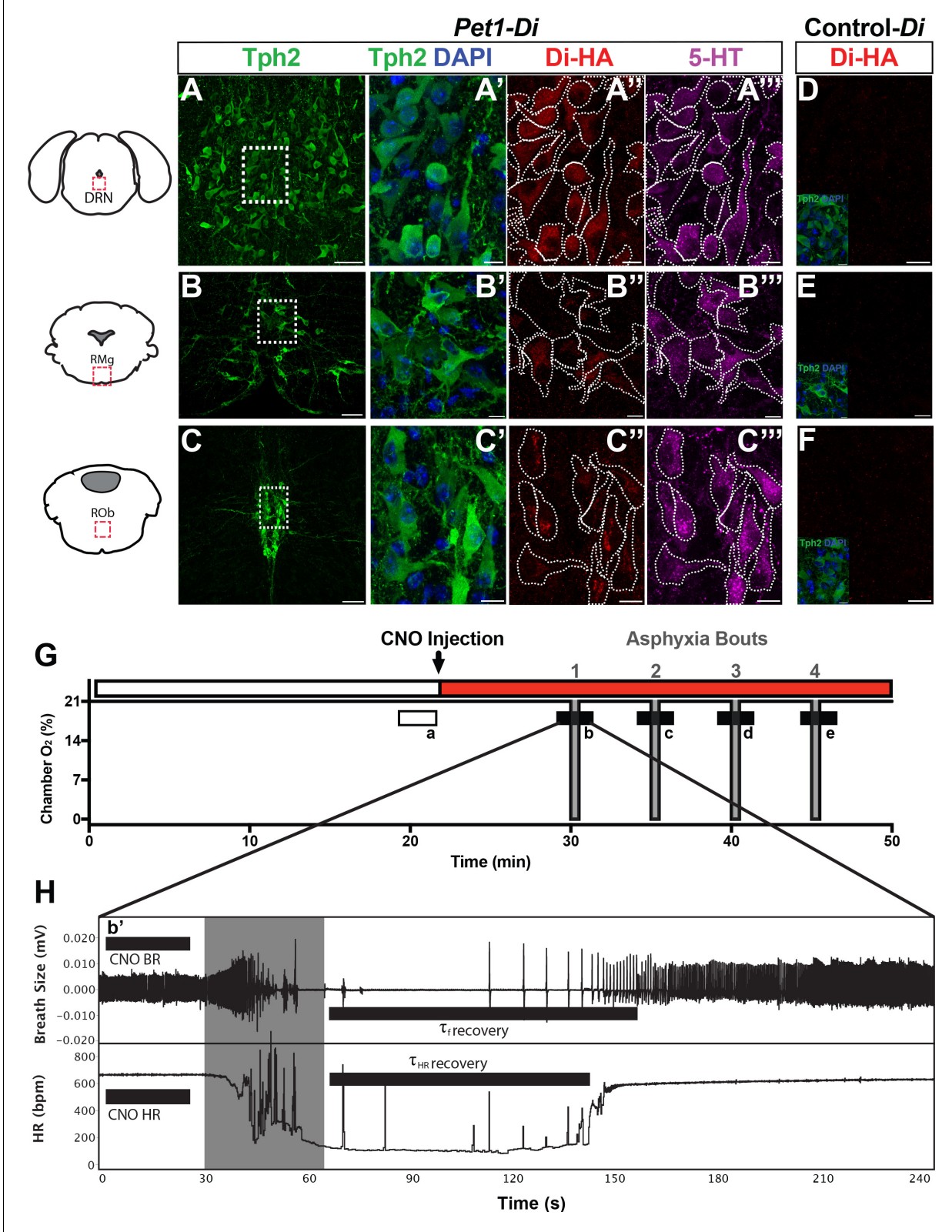

**Figure 1.** Genetic mouse model and postneonatal autoresuscitation assay. (A–C"') HA-tagged Di receptor expression targeted to *Pet1*-raphe serotonergic neurons in double-transgenic *Pet1-Flpe; RC-FDi* (referred to as *Pet1-Di*) pups at P8, as shown previously in adult mice (***Brust et al., 2014***). (A–C) Low magnification view of 20 μm coronal section showing neurons immunopositive for tryptophan hydroxylase 2 (Tph2), identifying serotonergic neurons in the dorsal raphe nucleus (DRN) (A), the raphe magnus nucleus (RMg) (B), and the raphe obscurus (ROb) (C). Fields demarcated by dashed

*Figure 1 continued on next page*

*Figure 1 continued*

rectangles in **A–C** are shown at higher magnification in **A'–C'"**, with Tph2 immunoreactivity again in green (**A'–C'**), HA-Di immunopositivity in red (**A"–C"**), serotonin (5-HT) immunopositivity in magenta (**A"'–C"'**), and dashed cell outlines as grossly determined by the Tph2 immunodetection signal. The raphe location, proportion, and intensity of HA-Di immunodetection signal was qualitatively similar across *Pet1-Di* pups from independent litters (***Figure 1—figure supplement 1***). (**D–F**) Representative fields from negative-control single transgenics harboring the unrecombined RC-FDi allele (referred to as Control-*Di*), showing no detectable HA-Di, in line with prior validation that Di-expression from RC-FDi requires Flpe-recombination. Insets show Tph2 immunodetection of serotonergic neurons in these fields. Scale bars in **A–C** equal 50 μm, and in **A'–C""** and D-F, 10 μm. (**G–H**) Schematic of repeated asphyxia-induced apnea and autoresuscitation recovery, during which breath size and heart rate are continuously monitored. (**G**) Plethysmograph chamber oxygen ($O_2$) percent across assay time, starting with ~20 min of pup acclimation to chamber air (21% $O_2$) including extraction of baseline cardiorespiratory values during the temporal window indicated by the open rectangle a. I.P. injection of CNO immediately follows; red rectangle indicates CNO exposure window. Asphyxia-apnea bouts are indicated by the four periods (b–e) of ~0% $O_2$ (97% $N_2$, 3% $CO_2$) shown in gray. (**H**) Temporally expanded view of an asphyxia-apnea bout including example tracing of breath size (and calculated breathing rate, BR) and heart rate (HR) over time. Primes (b' and similarly for c-e) indicate the bout-specific baseline post CNO injection immediately preceding asphyxia. The gray window indicates the period of asphyxia that induced the apnea, which is followed by immediate return to 21% $O_2$ during which autoresuscitation recovery ensues. Black bars after asphyxia indicate $\tau_f$ or $\tau_{HR}$ ($\tau$ defined as 63% of the baseline value before that specific asphyxic bout). Light gray box indicates period of asphyxia (97% N 3% $CO_2$) used to induce the apnea.

DOI: https://doi.org/10.7554/eLife.37857.003

The following figure supplement is available for figure 1:

**Figure supplement 1.** Additional examples of HA-Di expression in medullary brain tissue of *Pet1-Di* pups from independent litters.

DOI: https://doi.org/10.7554/eLife.37857.004

rectangles *b-e*), we used head-out plethysmography and ECG with continuous recording of breathing (frequency *f*, breaths • $min^{-1}$; and pressure changes associated with respiratory activity used to calculate tidal volume $V_T$, ml • $g^{-1}$), heart rate (HR), oxygen consumption ($\dot{V}_{O_2}$; ml • $min^{-1}$ • $g^{-1}$), and body temperature while maintaining pup thermoneutrality ($T_B$ at 36 ± 0.05°C) through chamber temperature adjustment ($T_A$ at 35–36 ± 0.05°C). From these measurements along with body mass (g), values were determined for minute ventilation ($\dot{V}_E$; ml • $min^{-1}$ • $g^{-1}$) and ventilatory equivalents for oxygen ($\dot{V}_E/\dot{V}_{O_2}$).

To assess if *Pet1* neurons at P8 modulate RA cardiorespiratory parameters, measurements were collected prior and during neuron perturbation (***Figure 1G***, open rectangle *a* versus filled rectangle *b'* of ***Figure 1H***, respectively). Initial baseline homeostatic characteristics showed no significant difference between *Pet1-Di* and control-*Di* pups, indicating that mere expression of Di in *Pet1*-lineage

**Table 1.** Baseline cardiorespiratory values prior to CNO-induced disruption of *Pet1*-neurons.
Data (mean ±standard deviation) for each time point were obtained prior to CNO-induced silencing (***Figure 1G*** open window *a*). Student's t-test was used to assess differences between genotypes.

| Mouse baseline characteristics | RC-FDi (Control-Di) n=15 | | *Pet1-Flpe*;RC-FDi (*Pet1-Di*) n=22 | | t-test p value |
|---|---|---|---|---|---|
| | Mean | SD | Mean | SD | |
| Weight (BW) (g) | 5.503 | 1.115 | 5.04 | 1.275 | 0.2616 |
| Breathing frequency (*f*) (breaths • $min^{-1}$) | 241.6 | 26.87 | 246.3 | 30.93 | 0.6406 |
| Tidal volume ($V_T$) (ml • $g^{-1}$) | 4.51 | 1.034 | 4.757 | 1.784 | 0.6322 |
| Minute ventilation ($\dot{V}_E$) (ml • $min^{-1}$ • $g^{-1}$) | 1070 | 169.9 | 1139 | 331.9 | 0.4618 |
| Heart rate (HR) (bpm) | 634.6 | 40.88 | 612.8 | 42.01 | 0.1271 |
| Oxygen consumption ($\dot{V}_{O_2}$) (ml • $min^{-1}$ • $g^{-1}$) | 77.43 | 19.19 | 71.25 | 14.39 | 0.2705 |
| Ventilatory equivalents ($\dot{V}_E/\dot{V}_{O_2}$) | 14.33 | 2.989 | 16.13 | 3.741 | 0.1281 |

DOI: https://doi.org/10.7554/eLife.37857.005

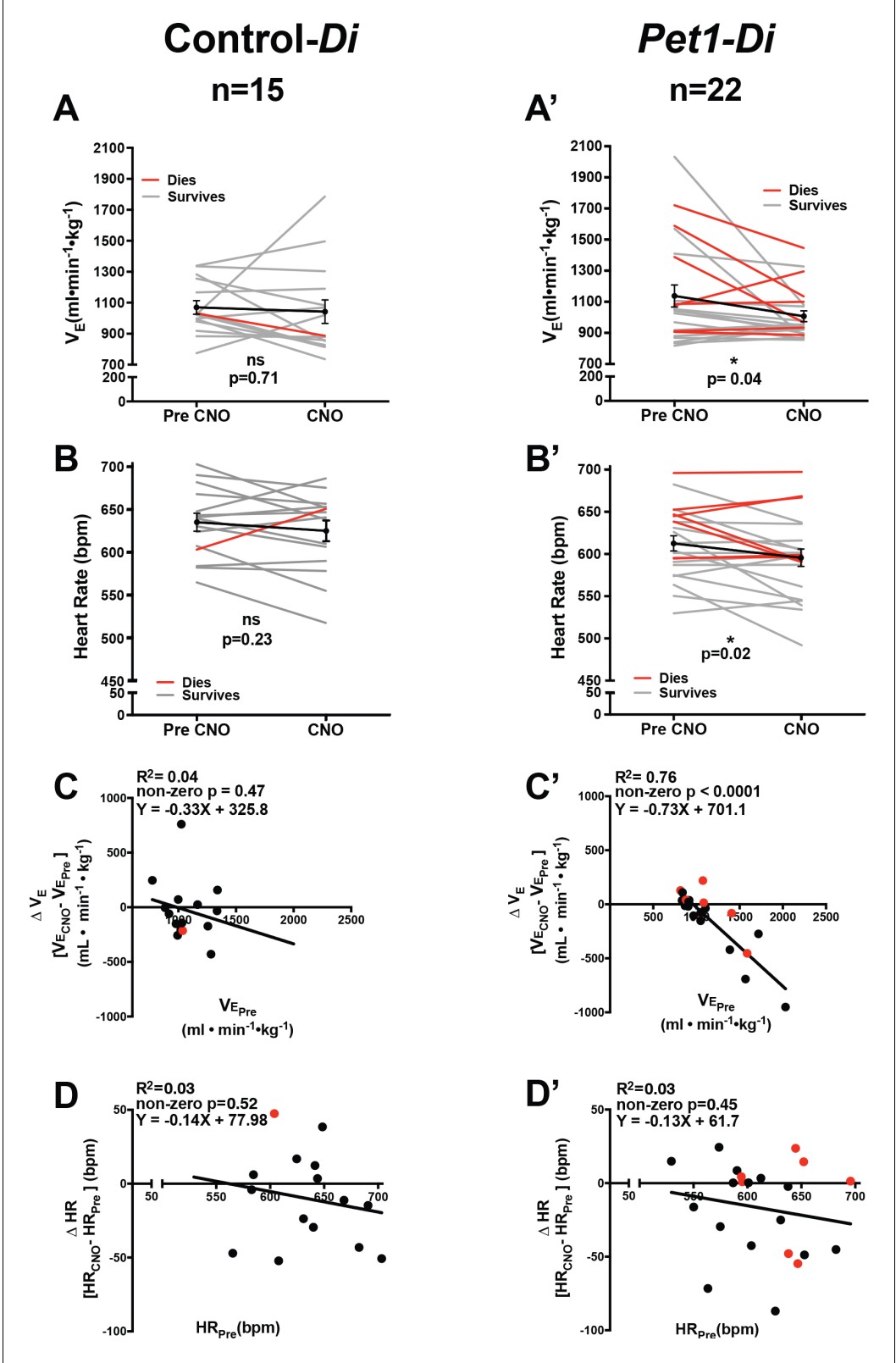

**Figure 2.** Acute perturbation of *Pet1* neurons alters baseline cardiorespiratory values. (**A–B'**) A and B represent minute ventilation ($\dot{V}_E$, ml min$^{-1}$ •kg$^{-1}$) and heart rate (HR, beats per minute, bpm), respectively, in sibling controls harboring the unrecombined RC-FDi allele (non-Di-expressors referred to as Control-*Di*) before and during CNO

*Figure 2 continued on next page*

*Figure 2 continued*

exposure. No detectable change was observed in $\dot{V}_E$ or HR before and during CNO exposure. (**A'** and **B'**) show $\dot{V}_E$ and HR, respectively, before and during CNO-induced disruption of *Pet1*-neurons in double transgenic *Pet1-Flpe*, RC-FDi (Di-expressors, referred to as *Pet1-Di*) pups. Coefficient-of-variation calculations for breathing *f* and HR during CNO exposure versus baseline prior to CNO injection are presented in ***Figure 2—figure supplement 1***. (**C–C'**) Linear regression of pre-CNO $\dot{V}_E$ plotted against change in $\dot{V}_E$ for control-*Di* pups (**C**) (correlation p=0.47 and $R^2$ = 0.04) and *Pet1-Di* pups (**C'**) (correlation p<0.0001 and $R^2$ = 0.76). (**D–D'**) Linear regression of pre-CNO HR plotted against change in HR for control-*Di* pups (**D**) (correlation p=0.52 and $R^2$ = 0.03) and *Pet1-Di* pups (**D'**) (correlation p=0.45 and $R^2$ = 0.03). Abbreviation $\dot{V}_{E\ Pre}$ (ventilation prior to CNO injection), $\dot{V}_{E\ CNO}$ (ventilation during CNO exposure), $HR_{Pre}$ (heart rate prior to CNO injection), $HR_{CNO}$ (heart rate during CNO exposure), bpm (beats per minute), mL (milliliters), min (minutes), kg (kilograms). 'Dies' refers to pups that go on to die in future bouts – red lines (**A–B'**) and red circles (**C–D'**), 'survives' refers to pups that survive the full set of asphyxic-apnea challenges – gray lines (**A–B'**) and black circles (**C–D'**). Linear regression of pre-CNO $\dot{V}_E$ plotted against change in HR, and change in $\dot{V}_E$ plotted against change in HR are presented in ***Figure 2—figure supplement 2***. Ventilatory equivalents for oxygen ($\dot{V}_E/\dot{V}_{O_2}$) and oxygen consumption $\dot{V}_{O_2}$ data pre- versus during CNO exposure are plotted in ***Figure 2—figure supplement 3***.

DOI: https://doi.org/10.7554/eLife.37857.006

The following figure supplements are available for figure 2:

**Figure supplement 1.** Variation in heart rate and respiratory rate.
DOI: https://doi.org/10.7554/eLife.37857.007
**Figure supplement 2.** Changes in baseline heart rate in relation to ventilation.
DOI: https://doi.org/10.7554/eLife.37857.008
**Figure supplement 3.** Analyses of ventilatory equivalents for oxygen and oxygen consumption before and during CNO exposure.
DOI: https://doi.org/10.7554/eLife.37857.009

cells (not yet triggered by CNO) as well as harboring and expression of the *Pet1-Flpe* transgene were neutral in this assay (***Table 1***).

Following CNO injection (***Figure 1H***, filled rectangle *b'*), double transgenic *Pet1-Di* pups (referred to as *Pet1-Di*-CNO) exhibited statistically significant decreases in $\dot{V}_E$ and HR (p=0.04 and p=0.02 respectively, ***Figure 2A' and B'***) not observed in control-Di pups (referred to as control-*Di*-CNO, ***Figure 2A and B***). Coefficient-of-variation calculations for breathing *f* and HR suggest comparable dispersion of the data obtained during CNO exposure as compared to baseline prior to CNO injection (***Figure 2—figure supplement 1***).

*Pet1-Di*-CNO pups with the highest $\dot{V}_E$ prior to CNO injection exhibited the largest $\dot{V}_E$ drop upon CNO administration and those with the lowest $\dot{V}_E$ prior to CNO exhibited modest, albeit not statistically significant, increases (***Figure 2A' and and C'***). This suggests that the ventilatory neurocircuitry may engage *Pet1* neurons to allow for greater deviation from a standard homeostatic set point, such that when *Pet1* neurons are inhibited the ventilatory dynamic range narrows overall. An alternative, more complex and arguably less likely technical explanation posits that pups with highest baseline $\dot{V}_E$ values are pups with highest Di expression levels (within the distribution determined by *R26/CAG* expression variation) such that CNO-triggering drives a greater cellular and circuit perturbation ultimately reflected in larger decreases in $\dot{V}_E$. However, HR findings do not lend support for this latter explanation, given that such a correlation was not present between the magnitude of HR changes upon CNO administration as compared to either pre-CNO HR, pre-CNO $\dot{V}_E$, or change in $\dot{V}_E$ (***Figure 2D'***, ***Figure 2—figure supplement 2***).

In contrast to these *Pet1-Di*-CNO-specific effects, no statistically significant effects were observed on ventilatory equivalents for oxygen ($\dot{V}_E/\dot{V}_{O_2}$) for either group (control-*Di*-CNO and *Pet1-Di*-CNO) (***Figure 2—figure supplement 3A,A',C,C'***). Both groups though showed a subtle decrease in oxygen consumption following CNO and return to the plethysmograph chamber (***Figure 2—supplement figure 3B, B'***).

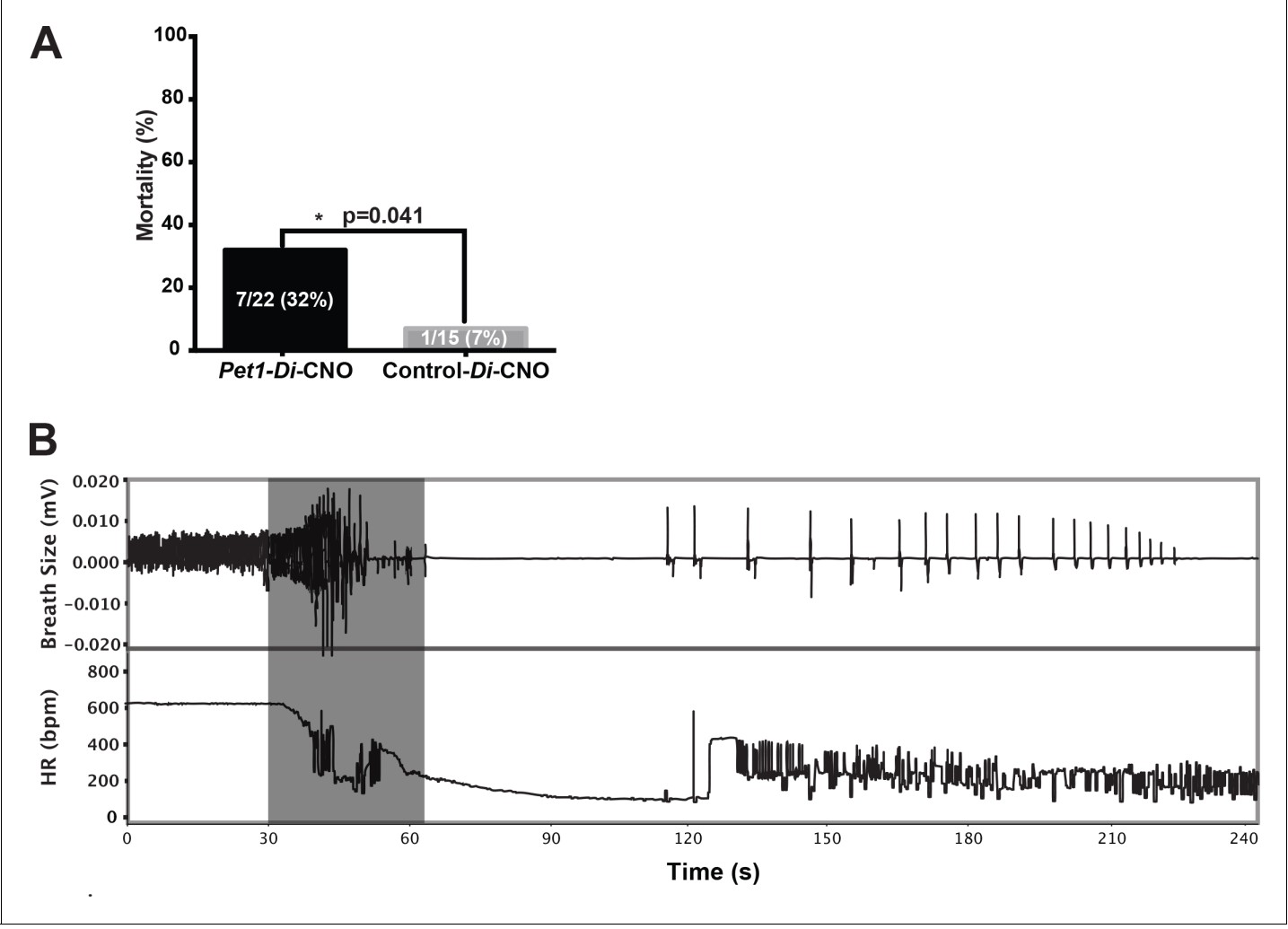

**Figure 3.** Acutely disrupting *Pet1*-neuron activity impairs pup ability to recover from multiple apneas. (A) Percent mortality after repeated asphyxia-induced apneas plotted as a function of genotype (one-sided Fisher's Exact Test with Lancaster's Mid P correction p=0.041). (B) Example breathing and heart rate response trace from a *Pet1-Di*-CNO pup that failed to recover. Gray box indicates the window of asphyxic conditions (97% N 3% CO₂). n = 15 for control-*Di*-CNO pups, n = 22 for *Pet1-Di*-CNO pups.

DOI: https://doi.org/10.7554/eLife.37857.010

## Impaired apnea recovery response and diminished pup survival following acute disruption of *Pet1*-neuron activity

We next queried whether CNO-Di-mediated disruption of *Pet1* neurons at P8 altered pup recovery from repeated, episodic, asphyxia-induced apneas (experimental paradigm schematized in *Figure 1G and H*, modified from previous studies (*Barrett et al., 2016*; *Cummings et al., 2011a*; *Erickson and Sposato, 2009*). We found autoresuscitation to be less effective in *Pet1-Di*-CNO pups in comparison with control-*Di*-CNO pups, resulting in a significant increase in mortality (*Figure 3A*, one-tailed Fisher Exact Test with Lancaster's Mid-P correction p=0.04). The calculated odds ratio for pup death as an outcome of asphyxic apnea in the face of *Pet1-Di*-CNO versus control-*Di*-CNO is 6.5, suggesting a substantially increased vulnerability to apneas when *Pet1* neuron activity is acutely perturbed. HR and breathing responses during a successful versus failed autoresuscitation are shown in *Figures 1H* and *3B*, respectively. In general, successful autoresuscitation is characterized by a brief primary apnea and rapid recovery of normal HR and eupneic breathing following the onset of gasping (*Figure 1H*), whereas failed autoresuscitation is characterized by a prolonged primary apnea (delayed gasping) and an inability to recover eupneic breathing despite gasping, and ultimately

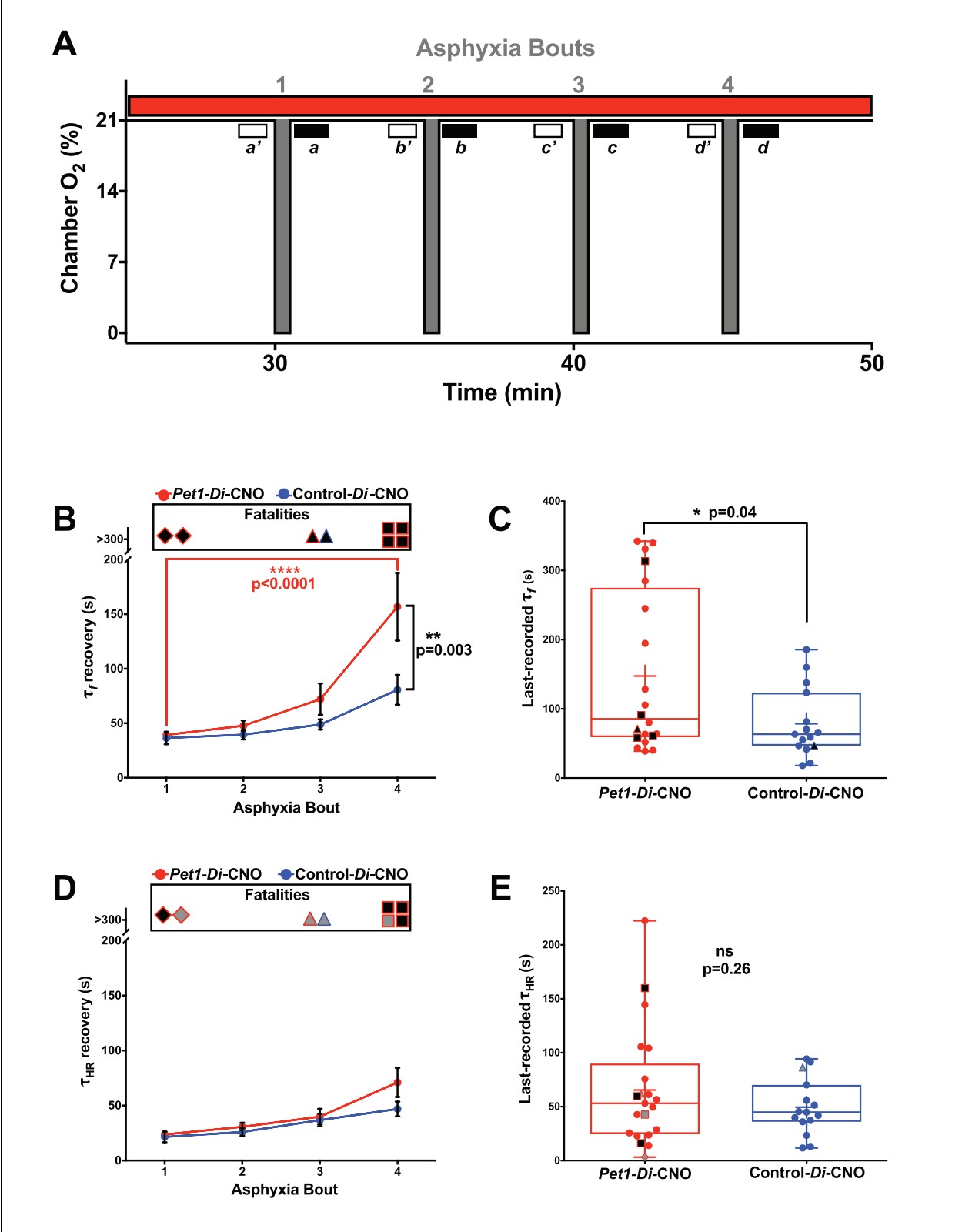

**Figure 4.** Acutely disrupting *Pet1*-neuron activity primarily affects respiratory components during apnea recovery  (A) Compressed schematic of autoresuscitation assay timeline indicating asphyxia-apnea bouts and data extraction windows; open windows *a'-d'* reflect bout-specific baseline measurements, filled windows *a-d* reflect temporally when recovery to 63% of baseline values was determined ($\tau_f$ and $\tau_{HR}$). (B and D) $\tau_f$ (B) or $\tau_{HR}$ (D) across bouts, with open 'Fatalities' rectangle above each graph denoting animals that died during the assay; the contained shapes represent the fatal

*Figure 4 continued on next page*

*Figure 4 continued*

apnea bout – diamond (bout 1), triangle (bout 3), square (bout 4) – while black fill reflects mortality during that recovery response. *Pet1-Di*-CNO pups indicated in red (n = 22), control-*Di*-CNO pups (sibling non-Di-expressors), in blue (n = 15). (**C and E**) $\tau_f$ (**C**) or $\tau_{HR}$ (**E**) during the last-recovered bout (final bout during which a pup was able to achieve 63% of baseline characteristic). Black-fill continues to reflect mortality, shape reflects bout that was fatal. Gray-filled shapes reflect animals that recovered HR to 63% of baseline although went on to die during that apnea bout despite meeting $\tau_{HR}$ criteria. Plotted circles reflect pups that survived all bouts; +indicates the mean; box-whisker plot shows median as the horizontal line, 1st and 3rd quartiles as the 'box,' and maximum and minimum values as the 'whiskers.'.

DOI: https://doi.org/10.7554/eLife.37857.011

failure to sustain HR at a recovered or near-recovered level (*Figure 3B*). Seven of twenty-two *Pet1-Di*-CNO pups compared to one of fifteen control-*Di*-CNO pups failed during the course of the assay to restore eupneic respiratory rhythm resulting in death (*Figure 3A*).

## Postneonatal *Pet1*-neuron disruption impaired breathing but not heart rate recovery in autoresuscitation

To query components of the autoresuscitation response and their possible dependency upon normal *Pet1*-neuron activity, we examined the recovery from each asphyxia bout (*Figure 4*), determining the recovery latency time ($\tau$) to achieve at least 3 s of breathing frequency *f* or HR at levels $\geq$ 63% of the eupneic *f* and HR observed as baseline immediately preceding the given apneic challenge (*Figure 4A,a'–d'*; *Figure 1G and H*). Time to 63% recovery was chosen for analysis because of the sensitivity likely offered via sampling that part of the recovery response which shows the largest amount of system change (system recovery), as predicted by the time constant ($\tau$) of a first-order, linear time-invariant system; while actual breathing and heart rate recovery systems may be more complex, sampling $\tau$ (as opposed to other time points) is our best prediction for maximally detecting recovery differences. Because not all pups survived the full 4-asphyxic-bout sequence (*Figure 4B and D*), we also separately analyzed characteristics of the last-survived bout for each pup (*Figure 4C and E*). Notably, there were some cases in which the fatal bout was nonetheless associated with a transient recovery of HR that met the $\tau$ conditions (HR $\geq$63% of pre-bout baseline sustained for $\geq$3 s), thus these values were used in calculating the mean $\tau_{HR}$ for the particular bout (*Figure 4D*, their inclusion being denoted by gray filled symbols at the top of the plot), and were included in the scatter plot of last-recorded $\tau_{HR}$ for each pup (*Figure 4E*, gray filled symbols). In contrast to HR, fatal bouts were never found associated with a transient recovery of breathing *f* that met the $\tau$ conditions (*f* $\geq$ 63% of pre-bout baseline sustained for $\geq$3 s).

Analysis of $\tau_f$ across asphyxia bouts showed a significant lengthening between the first and fourth bouts for *Pet1-Di*-CNO pups, but not for control-*Di*-CNO pups (*Figure 4B*, two-way ANOVA interaction p=0.027 *post hoc* Tukey's multiple comparison test p<0.0001 (*Pet1-Di*-CNO) and p=0.3063 (control-*Di*-CNO)). This $\tau_f$ prolongation associated with repeated apneic challenges in *Pet1-Di*-CNO pups, as calculated, is an underestimation given that animals that died (infinite attempted breathing rate recovery, if you will) could not be included. This further emphasizes the importance of *Pet1* neurons in enabling a rapid respiratory recovery response. Moreover, the mean $\tau_f$ in *Pet1-Di*-CNO pups during bout #4 was significantly prolonged by comparison to that of control-*Di*-CNO mice (*Figure 4B post hoc* Tukey's multiple comparison test p=0.003), indicating that, notwithstanding survival, the *Pet1-Di*-CNO pup response was abnormal.

In analyzing the last-recovered bout for all *Pet1-Di*-CNO pups, $\tau_f$ was again found to be prolonged (*Figure 4C*, p=0.04. Note that one of the surviving *Pet1-Di*-CNO pups, during bout 4, did not reach 63% of their pre-bout *f* within the allotted recording time (330 s) and thus was assigned a $\tau_f$ of 331 s – again, leading to an underestimation of the *Pet1-Di*-CNO effect and thus of the importance of *Pet*1 neuron activity to respiratory recovery.

Unlike $\tau_f$, $\tau_{HR}$ recovery did not show an interaction with asphyxia bout for *Pet1-Di*-CNO pups nor control-*Di*-CNO pups, (*Figure 4D*, two-way ANOVA interaction p=0.335). Additionally, in contrast to the strong effects on $\tau_f$ observed in *Pet1-Di*-CNO pups, the $\tau_{HR}$ was not significantly different from that of control-*Di*-CNO pups at any point during the assay, including the last-recovered bout (*Figure 4E* p=0.255). Of the seven *Pet1-Di*-CNO pups that died during the assay, three nonetheless reached 63% of their baseline HR during the fatal bout before succumbing to cardiac failure, and thus could be included in the $\tau_{HR}$ analysis for that bout (indicated as gray-filled symbols at the top of

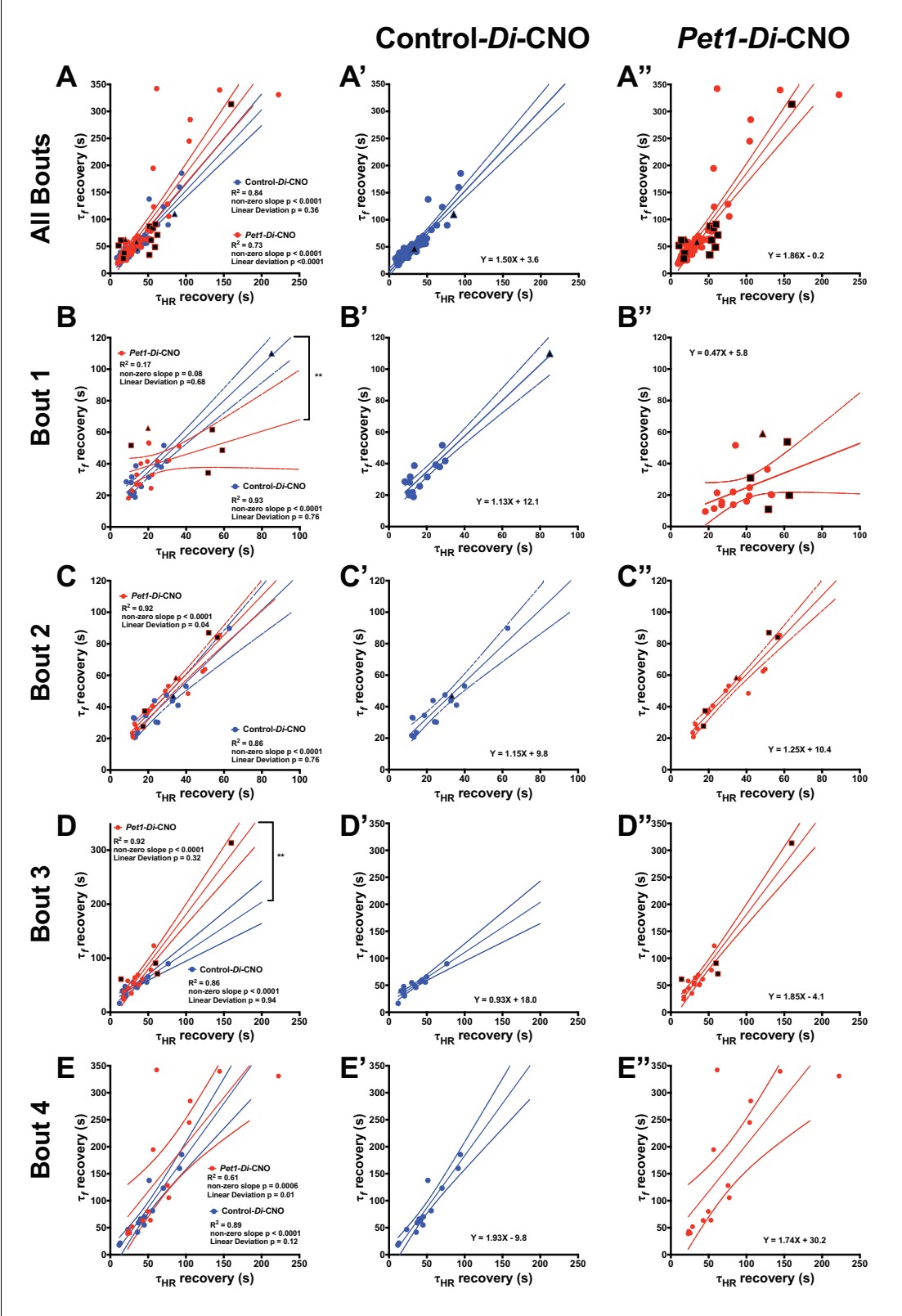

**Figure 5.** *Pet1-Di*-CNO pups show a decoupling of heart rate and breathing rate recovery times. (**A–A"**) Scatter plot of recovery $\tau_f$ (seconds, s) versus $\tau_{HR}$ (s) that includes all possible apnea recoveries of both groups, with data points from *Pet1-Di*-CNO pups in red, and *control-Di*-CNO pups in blue, superimposed for ease in comparing. For additional clarity, results from each group are presented separately in **A'**, control-Di-CNO; and **A"**, *Pet1-Di*-CNO. Similar scatter plots but only including recovery data from the first asphyxia-apnea bout (**B–B"**), from bout 2 (**C–C"**), bout 3 (**D–D"**), and bout 4

*Figure 5 continued on next page*

*Figure 5 continued*

(E–E"). Solid lines reflect a linear regression with dotted lines denoting 95% confidence intervals. Black filled shapes represent fatal bouts – diamond (bout 1), triangle (bout 3), square (bout 4). Circles reflect pups that survived all bouts. ** (p<0.01). Illustrative breathing and HR recovery tracings from a *Pet1-Di*-CNO pup and a Control-*Di*-CNO pup are presented in *Figure 5—figure supplement 1*.

DOI: https://doi.org/10.7554/eLife.37857.012

The following figure supplement is available for figure 5:

**Figure supplement 1.** Example traces showing differential breathing and HR recovery responses to asphyxia-induced apnea in a *Pet1-Di*-CNO pup (**A**) as compared to control (**B**).

DOI: https://doi.org/10.7554/eLife.37857.013

*Figure 4D* and in the scatter plot of 4E). The other four did not (indicated as black-filled symbols at the top of 4D and plotted in 4E), and thus their $\tau_{HR}$ values were necessarily excluded, again under-emphasizing the effect of *Pet1*-neuron function in HR recovery. The one control-*Di*-CNO pup that died also reached 63% of baseline HR during the terminal bout and thus was included in the $\tau_{HR}$ analyses (gray-filled, blue outlined symbols in *Figure 4D and E*).

Next, we plotted $\tau_f$ against $\tau_{HR}$ for each pup across all bouts to examine their relationship, given that *Pet1*-neuron silencing appeared to differentially affect breathing versus heart rate recovery. Applying a linear regression model to values generated from control-*Di*-CNO pups identified a non-random, linear relationship between $\tau_{HR}$ and $\tau_f$, whether analyzing all asphyxic bout recoveries (*Figure 5A* blue and extracted blue plot shown separately in 5A', Run's test linear deviation p=0.36, $R^2 = 0.84$, and non-zero slope p<0.0001) or the recovery response to each individual asphyxia bout (*Figure 5B–E* blue and extracted blue plot shown separately in 5B'−5E'). These control findings suggest that the cardiorespiratory response characteristics of P8 mouse pups interact in a direct, linearly correlated fashion, reflecting a well-coordinated breathing and heart rate recovery, likely important for maintaining adequate perfusion. By contrast in *Pet1-Di*-CNO pups, the linear nature of this relationship appears disordered (*Figure 5A* red and extracted red plot shown separately in 5A', Run's test linear deviation p<0.0001, $R^2 = 0.73$, and non-zero slope p<0.0001), with HR recovery proceeding in cases without the commensurate degree of breathing frequency *f* increases seen in controls. Appearing to drive a portion of these differences are the *Pet1-Di*-CNO pups that go on to die (black-filled red symbols). Owing to their future mortality, they drop out from subsequent bouts and as such are under sampled when all asphyxic bouts are analyzed. Given this, we also applied the linear regression model to each asphyxic-bout recovery response (*Figure 5B–E"*). Interestingly, during the initial asphyxia bout recovery, we found that the *Pet1-Di*-CNO pups have a weaker correlation between $\tau_{HR}$ and $\tau_f$ (*Figure 5B* red and extracted red plot shown separately in 5B' correlation p=0.08, $R^2 = 0.17$, linear deviation p=0.68), which differed from control-*Di*-CNO pups (*Figure 5B* blue and extracted blue plot shown separately in 5B' Run's test linear deviation p=0.76, $R^2 = 0.93$, and non-zero slope p<0.0001). This was further evidenced by the slope differences (*Figure 5B* p=0.001). Similar to the recovery analysis that includes all bouts, our analysis of just the first-bout recovery responses showed that the *Pet1-Di*-CNO pups that go on to die (*Figure 5B and B"*, black-filled symbols) had the greatest decoupling of HR and breathing *f* recovery kinetics. Interestingly, this decoupling is less apparent in the recovery response to bouts #2 and #3 but pronounced again in bout #4. Overall, perturbation of *Pet1* neurons may result in a decoupling of the cardiac and respiratory components central to a robust autoresuscitation response (sample illustrative tracings in *Figure 5—figure supplement 1*).

### Gasp response features of Pet1-Di-CNO pups track with mortality

Because *Pet1-Di*-CNO pups were less able to recover from repeated apneas, we sought to determine whether particular cardiorespiratory responses around an apnea tracked with later mortality. We examined the initial induced apnea (*Figure 4A* filled window *a*) so as to focus on characteristics independent of later size effects associated with repeated apneas. We found that *Pet1-Di*-CNO pups that died, by comparison to control-*Di*-CNO siblings who survived (14 of 15), exhibited a more disordered gasp response (*Figure 6A*) characterized by a smaller first gasp (*Figure 6B*, One-way ANOVA p=0.047, *post hoc* Tukey's multiple comparisons test p=0.04 for *Pet1-Di*-CNO dies and control-*Di*-CNO survives), a longer latency to that first gasp (*Figure 6C*, One-way ANOVA p=0.013, *post hoc* Tukey's multiple comparisons test p=0.02 for *Pet1-Di*-CNO survives versus *Pet1-Di*-CNO

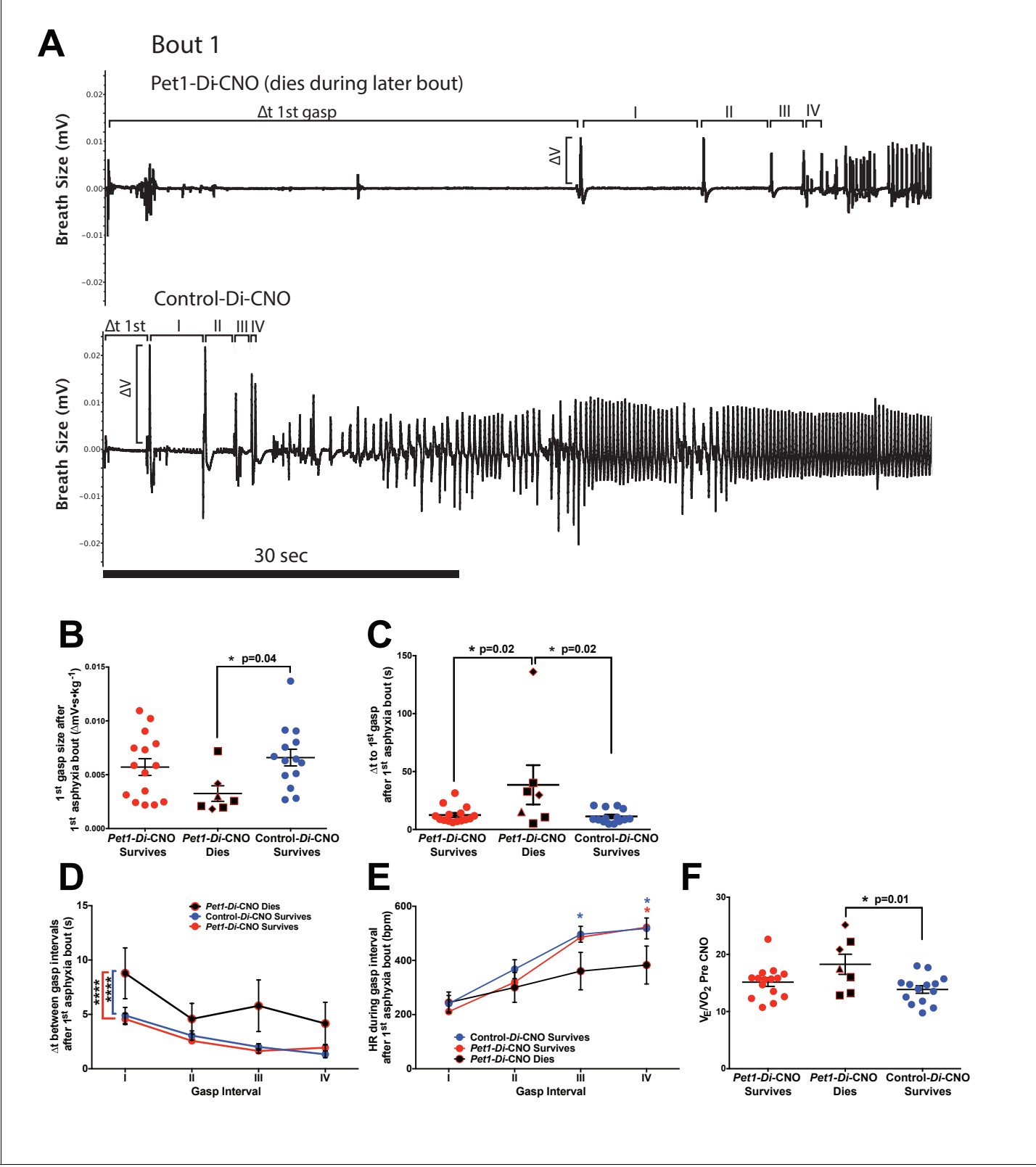

**Figure 6.** *Pet1-Di*-CNO pups demonstrate a disordered gasp response to the initial asphyxic bout and induced apnea. (A) Example respiratory traces immediately following the first asphyxia-induced apnea: top trace, from a *Pet1-Di*-CNO pup that goes on to die during a subsequent bout; bottom trace, from a control-*Di*-CNO pup (non-Di-expressor). Illustrated are differences in gasp characteristics and components that are plotted in B–D. (B) First gasp size (normalized to body weight). (C) Time to first gasp during initial apnea. (D) Time difference (s, seconds) between gasp intervals. (E) Mean

*Figure 6 continued on next page*

*Figure 6 continued*

heart rate (HR) between gasp intervals. (F) Baseline ventilatory equivalents per oxygen, $\dot{V}_E/\dot{V}_{O_2}$, prior to CNO administration in relation to genotype coupled with CNO exposure (neuron perturbation) and assay survival. Abbreviations: (I–IV) – time difference (s) between 1st and 2nd (I), 2nd and 3rd (II), 3rd and 4th (III), or 4th and 5th (IV) gasps, respectively. Red * or bracket compares *Pet1-Di*-CNO pups that go on to die versus *Pet1-Di*-CNO pups that survive. Blue * or bracket compares *Pet1-Di*-CNO pups that go on to die versus control-*Di*-CNO pups that survive. Shapes representing fatal bout – diamond (bout 1), triangle (bout 3), square (bout 4). Circles represent pups that survived all bouts. * (p<0.05) and **** (p<0.0001). Variability of baseline breathing and HR pre- versus during CNO exposure (prior to any asphyxic-apnea challenge) for *Pet1-Di* and control-*Di* pups is analyzed in *Figure 6—figure supplement 1* and *Figure 6—figure supplement 2*, respectively.

DOI: https://doi.org/10.7554/eLife.37857.014

The following figure supplements are available for figure 6:

**Figure supplement 1.** Variability for the baseline room air interbreath interval in Control-*Di* and *Pet1-Di* pups prior to and during CNO exposure.

DOI: https://doi.org/10.7554/eLife.37857.015

**Figure supplement 2.** Variability for the baseline room air interbeat interval in Control-*Di* and *Pet1-Di* pups prior to and during CNO exposure.

DOI: https://doi.org/10.7554/eLife.37857.016

dies, and p=0.02 for *Pet1-Di*-CNO dies versus control-*Di*-CNO), and prolonged inter-gasp intervals (*Figure 6D*, two-way ANOVA Genotype/survival p<0.0001, *post hoc* Tukey's multiple comparison test of means p<0.0001 and p<0.0001 for *Pet1-Di*-CNO dies versus control-*Di*-CNO survives, and *Pet1-Di*-CNO dies versus *Pet1-Di*-CNO survives, respectively) during the recovery from the first asphyxia-induced apnea. As an indirect measure of gasping effectiveness, we examined the characteristic transient increase in HR between each gasp that is required for homeostasis recovery. Here too we observed *Pet1-Di*-CNO pups who went on to die had a lower average HR between each gasp as the gasp train progressed compared to *Pet1-Di*-CNO surviving pups and control-*Di*-CNO pups (*Figure 6E*, two-way ANOVA genotype/survival p=0.012 and interaction p=0.043, *post hoc* Tukey's multiple comparison test p=0.042 and p=0.067 (gasp interval III) and p=0.044 and p=0.036 (gasp interval IV) for *Pet1-Di*-CNO dies versus control-*Di*-CNO survives, and *Pet1-Di*-CNO dies versus *Pet1-Di*-CNO survives, respectively). These findings suggest that the gasp characteristics of *Pet1-Di*-CNO pups that went on to die were not as effective at raising HR. Thus, disordered gasping after the first apnea bout associated with an increased risk for future mortality around subsequent asphyxic-apnea challenges.

## Homeostatic characteristics prior to acute *Pet1*-neuron perturbation and risk likelihood for mortality

To determine if particular baseline homeostatic characteristics increased the risk likelihood of future mortality when confronted by asphyxic challenge in the face of acute *Pet1*-neuron inhibition, we performed logistic regression analyses using as the independent variable either the baseline HR, $\dot{V}_E$, $\dot{V}_{O_2}$, or $\dot{V}_E/\dot{V}_{O_2}$ (data points obtained from the time point indicated by open window *a*, *Figure 1G*) and autoresuscitation outcome – death versus survival – as dependent variables. This approach would allow us to account for the state of the homeostatic network before acute perturbation. Of these input variables, only $\dot{V}_E/\dot{V}_{O_2}$ resulted in a statistically significant odds ratio (*Table 2*, p=0.027, odds ratio of 1.399), with higher values for $\dot{V}_E/\dot{V}_{O_2}$ correlating with increased risk of future death in

**Table 2.** Mortality by homeostatic characteristics prior to neuron perturbation.
Logistic regression results of different homeostatic characteristics and mortality outcomes of *Pet1-Di*-CNO pups.

| Baseline characteristic | Odds ratio | Confidence interval | | p-value* |
|---|---|---|---|---|
| Heart rate | 1.017 | 0.994 | 1.040 | 0.15 |
| Ventilation | 1.001 | 0.999 | 1.004 | 0.38 |
| Maximum Oxygen Consumption | 0.965 | 0.907 | 1.026 | 0.25 |
| Ventilatory Equivalents | 1.399 | 1.039 | 1.883 | 0.027 |

*logistic regression, controlling for genotype

DOI: https://doi.org/10.7554/eLife.37857.017

*Pet1-Di*-CNO pups. No differences were found between $\dot{V}_E/\dot{V}_{O_2}$ mean and variance values between *Pet1-Di*-CNO and contro*l-Di*-CNO pups (p=0.128 and p=0.392, respectively). When *Pet1-Di*-CNO pups are separated by mortality, we similarly observe that *Pet1-Di*-CNO pups that go on to die have a significantly higher $\dot{V}_E/\dot{V}_{O_2}$ when compared to control-*Di*-CNO pups who survive (*Figure 6F*, one-way ANOVA p=0.018 with Tukey's multiple comparisons test p=0.01). Thus, inhibition of *Pet1* neurons in mouse pups whose baseline $\dot{V}_E/\dot{V}_{O_2}$ value resides at the higher end of the $\dot{V}_E/\dot{V}_{O_2}$ distribution, increases the probability that they will go on to die when confronted by repeated asphyxia-induced apneas.

We also explored variability of baseline breathing (*Figure 6—figure supplement 1*) and HR (*Figure 6—figure supplement 2*) pre- versus during CNO exposure (prior to any asphyxic-apnea challenge) for *Pet1-Di* and control-*Di* pups by evaluating the standard deviation (SD) of the interbreath and interbeat intervals, the SDxSD axis of Poincare first return plot for interbreath interval and interbeat interval, and the root mean square of successive differences (RMSSD). Significant findings were limited to interbreath interval parameters, specifically for *Pet1-Di* pups pre- versus during-CNO exposure that would go on to die in the assay (*Figure 6—figure supplement 1*), with CNO exposure (and thus *Pet1* neuron perturbation) associating with a decrease in SD and decrease in estimated area of the Poincare first return plot, both suggesting a decrease in interbreath interval variability in the time domain.

## Discussion

### Strategy

Cardiorespiratory homeostasis involves central and peripheral neural circuits working in concert to sense and respond to tissue conditions of hypercapnia, hypoxia, and acidosis. These circuits in neonates are newly engaged for *ex utero* life, including recovery from apneas, which occur more frequently during infancy. Based on recent genetic mouse models implicating *Pet1*-expressing serotonergic neurons in cardiorespiratory homeostasis (*Barrett et al., 2016*; *Brust et al., 2014*; *Cummings et al., 2011a*; *Cummings et al., 2013*; *Erickson and Sposato, 2009*; *Erickson et al., 2007*; *Ray et al., 2011*), we hypothesized that in neonates too *Pet1* neurons play an important, real-time role, including in the recovery response to apneas. To test this free of ambiguity associated with chronic, developmental perturbations and potentially hidden compensatory events (*Barrett et al., 2016*; *Erickson et al., 2007*; *Erickson and Sposato, 2009*), we applied an inducible, acute neuronal inhibition strategy involving targeted expression of hM4Di in *Pet1* neurons, as achieved in double transgenic *Pet1-Flpe, RC-FDi* pups, with the cognate ligand CNO injected intraperitoneally at P8 to trigger *Pet1*-neuronal inhibition. In the presence or absence of this acute perturbation, we assayed cardiorespiratory function at baseline and during apnea induction and recovery, the latter allowing for exploration of the gasp response and the ability to rapidly restore HR and eupneic breathing.

### Main findings

Significant findings include the following: (1) Repeated asphyxia-induced apneas during CNO exposure resulted in a greater frequency of failed autoresuscitation in *Pet1-Di*-CNO pups (7 of 22 Di-expressing pups) as compared to control-*Di*-CNO pups (1 of 15 non-Di-expressing, RC-FDi-harboring pups). (2) Baseline room air cardiorespiratory function ($\dot{V}_E$ and HR) was modestly but statistically significantly altered following acute, CNO-Di-mediated perturbation of *Pet1* neurons in P8 pups (*Pet1-Di*-CNO pups); this contrasts with CNO-treated sibling controls (control-*Di*-CNO pups) which showed no detectable changes in these baseline properties. Findings suggest that *Pet1* neurons may normally enable greater cardiorespiratory dynamic range, which narrows upon *Pet1*-neuron inhibition. (3) *Pet1-Di*-CNO pups during their last recovered asphyxic apnea-inducing bout, whether the assay-concluding fourth bout (for pups that survived) or the bout just prior to the fatal failed bout, took significantly longer to recover to 63% of the pre-bout baseline eupneic breathing as compared to controls. (4) In contrast to the impaired respiratory recovery characterizing *Pet1-Di*-CNO pups, the time to recover HR to 63% of the pre-bout baseline was indistinguishable from that of control-*Di*-CNO pups. (5) A linear relationship between HR and breathing *f* recovery was observed in the autoresuscitation response of control-*Di*-CNO pups, but was decoupled in *Pet1-Di*-CNO pups. (6)

The gasp response to the initial, survived apneic challenge was disordered in the *Pet1-Di*-CNO pups that would go on to die during a subsequent apnea; the first gasp was smaller, the latency to first gasp longer, inter-gasp intervals prolonged, and the HR increase became smaller as the gasp train progressed. (7) Pups exhibiting modest hyperventilation – within the high end of the distribution prior to any neuronal perturbation – had a higher risk likelihood for autoresuscitation failure when subjected to the combined stressors of acute *Pet1*-neuron inhibition and apneic challenge.

## *Pet1* neurons shape the neonatal P8 cardiorespiratory homeostatic set point and dynamic range

Here we provide evidence through selective and, importantly, acute neuronal perturbation to support the hypothesis that *Pet1* neurons at P8 play an active role in shaping the neonatal cardiorespiratory homeostatic set point and the capacity to mount a robust autoresuscitation response. We first explored baseline cardiorespiratory properties of P8 mouse pups and whether they changed following acute *Pet1* neuron perturbation. The starting cardiorespiratory values were indistinguishable between *Pet1-Di*-expressing pups (*Pet1-Flpe*, RC-FDi double transgenics) and sibling controls (RC-FDi single transgenics) (*Table 1*) suggesting relative neutrality around expression of the untriggered Di receptor in *Pet1* neurons. Upon CNO administration, Di-expressing double transgenics exhibited cardiorespiratory changes, specifically an overall decrease, albeit subtle, in $\dot{V}_E$ and HR (*Figure 2A,A' and B,B'*, respectively). This is consistent with *Pet1* neurons playing an active role in neonates in maintaining both respiratory and cardiac tone.

Prior studies involving chronic, developmental disruption of *Pet1* neurons showed only diminished HR by P8, with $\dot{V}_E$ levels indistinguishable from the control cohort (*Barrett et al., 2016*; *Cummings et al., 2011a*; *Cummings et al., 2013*). This lack of detectable $\dot{V}_E$ effect under conditions of chronic *Pet1*-neuron perturbation could reflect compensatory circuit plasticity around ventilation, but which occurs to a lesser extent around HR control. It is also possible that the inducible, acute neuronal perturbation approach offers greater sensitivity and thus capacity to uncover more extensive phenotypes: as applied here, it allowed each animal to serve as its own control, enabling within-animal comparisons across pre- versus during-perturbation measurements, minimizing between-animal variability. An additional benefit of the inducible-perturbation approach is that body weight variation among pups was negligible, lessening technical variability associated with acquiring plethysmographic measurements on especially small pups; by contrast, chronic developmental perturbations of *Pet1* neurons results in impaired growth and diminished body weight (*Barrett et al., 2016*; *Cummings et al., 2011a*; *Cummings et al., 2013*; *Erickson et al., 2007*; *Pelosi et al., 2014*; *Yang and Cummings, 2013*).

Also uniquely uncovered by employing an inducible manipulation approach was the finding that the $\dot{V}_E$ state of the animal before manipulation tracked with the size of the $\dot{V}_E$ change during perturbation. Furthermore, we observed that perturbation of *Pet1* neuron activity resulted in a regression toward a common $\dot{V}_E$ set point. This may indicate that *Pet1* neuron activity is an important component that allows the internal arousal state of the animal to alter $V_E$ in preparation for a stressor requiring higher ventilation. Collectively, our findings of decreased HR and $\dot{V}_E$ immediately following CNO-Di-mediated perturbation of *Pet1* neurons provides evidence that, even without an external stressor like asphyxia-induced apnea or exposure to $CO_2$, neonatal mice use *Pet1* neurons to shape a homeostatic set point.

Baseline $\dot{V}_{O_2}$ measurements were less straightforward, showing in both genotypes a subtle decrease following CNO and return to the plethysmograph chamber. We speculate that this non-specific $\dot{V}_{O_2}$ effect reflects relaxation in and habituation to the chamber at this advancing time point in the assay resulting in a subtle lowering of metabolic rate. It could also reflect the very real challenge in accurately measuring $\dot{V}_{O_2}$ (as compared to the other cardiorespiratory parameters) for such tiny mouse pups and/or reflect a modest effect of CNO itself (independent of Di expression [*Gomez et al., 2017*; *Manvich et al., 2018*]) on P8 pup metabolic rate.

## Acute perturbation of *Pet1* neurons disrupts the normal apnea response in neonates

Not only were baseline cardiorespiratory properties affected in P8 neonates upon acute disruption of *Pet1*-neuron activity, but also and more strikingly the capacity to autoresuscitate from repeated

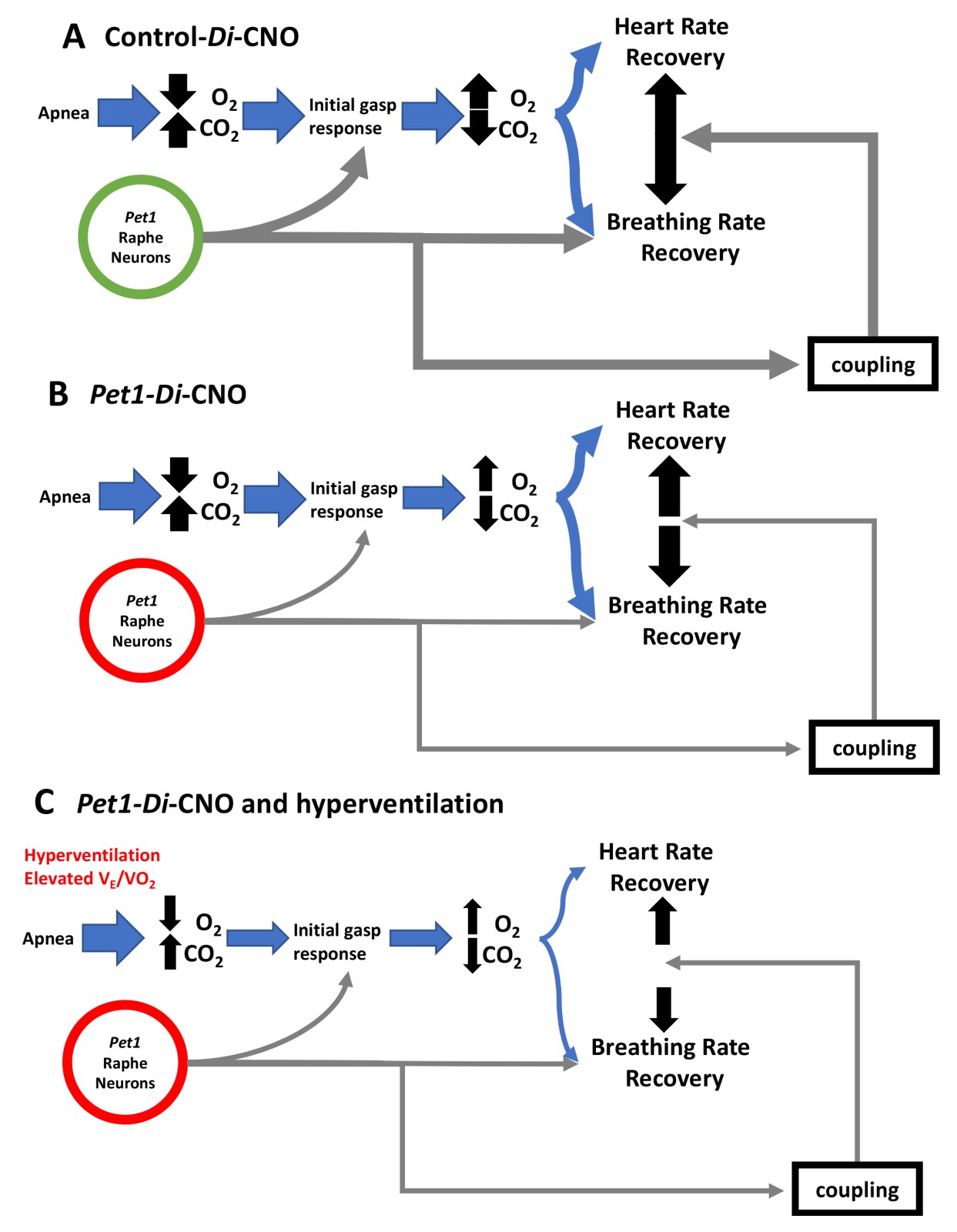

**Figure 7.** Proposed *Pet1*-neuron involvement in the postneonatal autoresuscitation response to apneas. (**A**) In control pups (unrecombined RC-FDi allele harboring, non-Di expressors exposed to CNO), an apnea (with associated bradycardia) would lead to blood/tissue oxygen tension ($PO_2$) reduction and carbon dioxide tension ($PCO_2$) elevation, which in turn would stimulate *Pet1* raphe neuron activity along with other brainstem areas to trigger gasping, resulting in homeostatic $PO_2$ elevation and $PCO_2$ reduction enabling heart and breathing rate recovery, which occurs in a linearly

*Figure 7 continued on next page*

*Figure 7 continued*

coupled fashion, ultimately restoring life-supporting $PO_2$ and $PCO_2$ levels. (**B**) In *Pet1-Di*-CNO pups in which *Pet1* neurons are acutely perturbed, the initial gasp response is disordered and diminished, as is the breathing rate recovery, which becomes decoupled from the more normal heart rate recovery trajectory. Collectively this leads to prolonged and potentially life-threatening $PO_2$ and $PCO_2$ levels. (**C**) Worsening yet is when spontaneous conditions of hyperventilation (elevated $\dot{V}_E/\dot{V}_{O_2}$) couple with *Pet1* neuron perturbation. The dual insult results in diminished collective drive across *Pet1* and other neuron types, resulting in greater impairment in gasping and a more severe decoupling of the breathing and heart rate recovery responses, increasing significantly the risk likelihood of death.

DOI: https://doi.org/10.7554/eLife.37857.018

asphyxia-induced apneas. Mortality was increased significantly and autoresuscitation prolonged across the assay chain of apneic challenges. Thus, the neonatal homeostatic recovery response appears to actively require *Pet1*-neuron function for life-supporting robustness, with death in ~32% of the *Pet1-Di*-CNO pups as compared to ~7% in control-*Di*-CNO pups. Prior chronic developmental perturbation studies (*Barrett et al., 2016*; *Cummings et al., 2011a*), while resulting in similar increases in mortality at P8, were unable to resolve temporally whether *Pet1*-neuron activity was required during embryogenesis for the broader establishment of autoresuscitation circuitry or actually required at P8 as a fundamental participant in the homeostatic response. Present findings indicate active participation at P8, revealing an active component in neonatal homeostatic circuitry and informing possible intervention inroads for mitigating risk of autoresuscitation failure.

An additional striking finding in *Pet1-Di*-CNO pups was that deficits in the respiratory recovery response alone, without similarly early and severe dysfunction in cardiac recovery, were sufficient to increase mortality after an apnea (*Figures 4* and *5*). This decoupling of the heart rate recovery response and the respiratory recovery response suggests segregation in the underlying circuitry and its modulation by *Pet1* neurons at P8, building upon the segregation of phenotypes between chronic versus acute *Pet1*-neuron perturbation described above. Disruption of this link between respiratory and cardiac responses likely contributes to the increased mortality. Notably, continued HR elevation in the unsupportive setting of unresolved hypoxia has been shown in other models to increase mortality (*Scremin et al., 1980*); similar conditions may arise in *Pet1-Di*-CNO pups when productive gasping and ventilation recovery lags behind and out of sync with the HR recovery.

## Initial response characteristics correlate with future apnea-related mortality

While autoresuscitation failure or prolongation ultimately characterized most *Pet1-Di*-CNO pups, in the initial asphyxia-apnea bout nearly all recovered, which allowed us to ask if there were features of that first recovery response that were specific to pups that ultimately died during one of the subsequent apnea challenges. Indeed, we found discrete measurable differences in the gasping response to the first apnea challenge, as compared to controls: a smaller first gasp, a prolonged latency to first gasp, longer inter-gasp intervals. One interpretation is that an initial suboptimal gasp response, with the associated extended conditions of poor oxygenation despite ultimate apnea recovery, may set in motion some cellular deficiency that then predisposes to mortality during a forthcoming apnea. Further, our findings indicate that these impaired gasps are unable to trigger the same type of heart rate recovery during the initial apnea-recovery attempt even though the overall timing to recover 63% of HR is not impaired. We suggest that the initial suboptimal gasp response may be indicative of an intrinsic failing of the broader respiratory response, not just the gasp response, with this broader failing the result of acute *Pet1*-neuron perturbation, which ultimately increases mortality risk upon exposure to asphyxic-apnea stress. Given the high frequency of apneas among human infants (*Daily et al., 1969*; *Kelly et al., 1985*; *Southall et al., 1980*), measuring apnea recovery (gasp size, timing, HR between gasps, and breathing and heart rate coupling) could be a potential indicator of raphe neuron dysfunction and a need for targeted infant monitoring during the peak age for SIDS risk of 2–4 months.

In addition to identifying initial gasp response features that distinguished pups that died on exposure to multiple apneas, we also identified the homeostatic state of modest hyperventilation – $\dot{V}_E/\dot{V}_{O_2}$ at the higher end of the distribution – as increasing the risk likelihood of death in pups challenged with the combined stressors of acute *Pet1*-neuron dysfunction and repeated apneas. When

at the high end of the $\dot{V}_E/\dot{V}_{O_2}$ distribution, the normally strong drivers of gasping, heart rate, and breathing rate recovery – i.e. increased $PCO_2$ and decreased $PO_2$ – are countered because these pups at baseline are always in a state of modest overbreathing and thus relatively hypocapnic and hyperoxic such that more severe apneic conditions would be required to trigger the respiratory response. This then could contribute to the insufficient gasp response, which despite the pup starting out with an elevated $\dot{V}_E/\dot{V}_{O_2}$ could ultimately lead to more severe hypoxemia due to HR and breathing rate decoupling in the setting of *Pet1*-neuron dysfunction, resulting in a decreased ability to withstand future apneas (*Figure 7*). In control, wild-type pups, redundancy in the respiratory response may permit sufficiently robust homeostatic responses despite such modest hyperventilation so as to withstand repeated apneas; however, if simultaneous with *Pet1*-neuron dysfunction, it may no longer be possible to mount the necessary life-sustaining response (*Figure 7C*). Such vulnerability may be exacerbated during active versus quiet sleep, as recently suggested by studies in *Tph2*$^{-/-}$ rat pups (*Magnusson and Cummings, 2018*; *Young et al., 2017*). As well, it is worth noting that medullary *Pet1* neurons project to brainstem centers relevant to cardiorespiratory control, such as the pre-Bötzinger complex with its role in respiratory rhythm and chemosensory processing, the nucleus ambiguous housing cardiovagal neurons, and the nucleus of the solitary tract, an important site of integration of the chemoreflex (peripheral and central) (*Brust et al., 2014*; *Feldman et al., 2003*; *Hennessy et al., 2017*; *Hodges and Richerson, 2010*; *Wang and Richerson, 1999*).

## Conclusions

In summary, we found that *Pet1* neurons play an active role in P8 mouse pups in maintaining cardiorespiratory tone and in supporting robust, life-sustaining autoresuscitation responses to asphyxia-induced apneas. When *Pet1* neurons were compromised acutely, $\dot{V}_E$ decreased, HR slowed, and autoresuscitation failure increased. Respiratory more than cardiac recovery was impaired, causing a disorganization of the normal linear relationship between breathing *f* and HR. Moreover, early gasping abnormalities distinguished the *Pet1*-neuron-compromised pups that went on to die during subsequent apneas, as did modest baseline hyperventilation. Collectively, these findings shed new light on cardiorespiratory control systems and, more specifically, support a potential pathoetiological role for the SIDS-associated finding of postmortem brainstem 5-HT neuron abnormalities. Further, they suggest that gasp features might potentially help define a physiological profile associated with a higher risk likelihood for SIDS.

# Materials and methods

## Key resources table

| Reagent type (species) or resource | Designation | Source or reference | Identifiers | Additional information |
|---|---|---|---|---|
| Genetic reagent (*M. musculus*) | *Pet1-Flpe* | PMID: 18344997 | | Dr. Susan Dymecki (Department of Genetics, Harvard Medical School) |
| Genetic reagent (*M. musculus*) | RC-FDi Gt(ROSA)26Sor$^{tm(CAG-FSF-CHRM4*(Di))Dym}$ | PMID: 21798952 | derivative of MGI:5790683 Gt(ROSA)26Sor$^{tm9(CAG-mCherry,-CHRM4*)Dym}$ | Dr. Susan Dymecki (Department of Genetics, Harvard Medical School) |
| Antibody | Rabbit polyclonal anti-Tph2 | Novus Biological | Cat. #: NB100-74555 RRID:AB_1049988 | IHC (1:1000) |
| Antibody | Rat monoclonal anti-HA | Chromotek | Cat. # 7C9 RRID: AB_2631399 | IHC (1:200) |
| Antibody | Goat polyclonal anti-5-HT | Abcam | Cat. # ab66047 RRID:AB_1142794 | IHC (1:1000) |
| Antibody | Donkey anti-rabbit IgG-Alexa Fluor 488 | ThermoFisher Scientific | Cat. # A-21206 RRID:AB_2535792 | IHC (1:500) |
| Antibody | Donkey anti-rat IgG-Alexa Fluor 594 | ThermoFisher Scientific | Cat. # A-21209 RRID:AB_2535795 | IHC (1:500) |
| Antibody | Donkey anti-goat igG-Alexa Fluor 647 | ThermoFisher Scientific | Cat. # A-21447 RRID:AB_2535864 | IHC (1:500) |

*Continued on next page*

*Continued*

| Reagent type (species) or resource | Designation | Source or reference | Identifiers | Additional information |
|---|---|---|---|---|
| Recombinant DNA reagent | Flpe forward primer | 5'-GCATCTGGGAGATC ACTGAG-3' | | PCR genotyping |
| Recombinant DNA reagent | Flpe reverse primer | 5'-CCCATTCCATGCGG GGTATCG-3' | | PCR genotyping |
| Recombinant DNA reagent | FDi forward primer | 5'-CGAATTCGGAAACATA ACTTCG-3' | | PCR genotyping |
| Recombinant DNA reagent | FDi reverse primer | 5'-GGCAATGAAGACTTT CCACCG-3' | | PCR genotyping |
| Chemical compound, drug | clozapine-N-oxide (CNO) | Sigma | Cat. # C0832 | fresh stock solution 1 mg/ml in saline |
| Chemical compound, drug | DAPI (4', 6-diamidino-2-phenylindole) | ThermoFisher Scientific | Cat. # D1306 | nuclear counter staining, final concentraion at 5 µg/mL |

## Ethical approval

All experimental protocols were approved at Harvard Medical School (HMS) and the Geisel School of Medicine at Dartmouth by the respective Institutional Animal Care and Use Committees (IS00000231-3 and 2035, respectively) and the HMS Committee on Microbiological Safety (15-225), and were in accordance with the animal care guidelines of the National Institutes of Health.

## Experimental animals

For acute, chemogenetic perturbation of *Pet1* neurons in vivo, double transgenic mouse pups of the genotype *Pet1-Flpe,* RC-FDi (referred genotypically as *Pet1-Di*) were generated via crossing *Gt (ROSA)26Sor^{tm(CAG-FSF-CHRM4*)Dym}* (denoted in short-hand as RC-FDi [*Ray et al., 2011*]) homozygous females to hemizygous *Pet1-Flpe* (*Jensen et al., 2008*) males. Here Flpe expression mediates *FRT* recombination of the RC-FDi allele resulting in expression of the inhibitory, synthetic G protein-coupled receptor Di exclusively in neurons with current or a lineal history of *Pet1* expression, thus enabling acute, CNO-triggered Di inhibition of serotonergic neurons upon CNO i.p. injection. We refer to these CNO-treated double transgenic mice as *Pet1-Di-CNO* mice. Littermate controls were of the single transgenic RC-FDi genotype and thus devoid of Di expression but of comparable genetic background (predominantly C57BL/6J, minor 129) thus serving as controls (CNO-treated controls referred as *control-CNO*). Mice were obtained from nine independent litters, yielding 22 double transgenic *Pet1-Flpe,* RC-FDi pups (11 males, 11 females) and 15 single transgenic RC-FDi pups (five males, 10 females). Mice were housed in a temperature-controlled environment on a 12:12 hr light-dark cycle in an external housing environment with *ad libitum* access to standard rodent chow and water. Past experiments with similar genotypes and physiological measures demonstrated that a group size of n $\geq$ 15 would provide sufficient statistical power to detect differences between experimental and control groups (*Barrett et al., 2016*; *Brust et al., 2014*).

## Genotyping

Genotypes were determined as previously described (*Brust et al., 2014*). Briefly, DNA isolates from tail tip biopsies from P3-5 pups were subjected to PCR amplification using Taq DNA polymerase (New England BioLabs Inc.) and the following primer sequences (Invitrogen, Carlsbad, CA) diagnostic for *Flpe* (800 bp amplicon) or *hM4Di* (268 bp amplicon): 5'-GCATCTGGGAGATCACTGAG-3' (*Flpe* forward primer); 5'-CCCATTCCATGCGGGGTATCG-3' (*Flpe* reverse primer); 5'-CGAA TTCGGAAACATAACTTCG-3' (*FDi* forward primer); 5'-GGCAATGAAGACTTTCCACCG-3' (*FDi* reverse primer). PCR amplification consisted of an initial 5 min denaturation at 94 ℃, followed by 35 cycles, each consisting of 1 min at 94 ℃, 1.5 min at 60 ℃, and 1 min at 72 ℃, followed by a final 10 min extension at 72 ℃.

## Immunohistochemistry

For preparation of double transgenic *Pet1-Flpe,* RC-FDi tissue, postnatal day eight mice were briefly anesthetized with ice and immediately perfused intracardially with phosphate buffered saline (PBS)

followed by 4% paraformaldehyde (PFA) in PBS. Brains were extracted, soak-fixed for 2 hr in 4% PFA at 4°C, cryoprotected in 30% sucrose/PBS for 48 hr, and subsequently embedded in OCT compound (Tissue-Tek). Coronal sections were cryosectioned at 20 µm and mounted onto glass slides, were then rinsed three times with PBS for 10 min and permeabilized with 0.5% Triton X-100 in PBS for 1 hr, and blocked in 5% normal donkey serum (NDS, Jackson ImmunoResearch), 1% BSA, 0.5% Triton X-100 in PBS for 1 hr at room temperature (RT). Sections were rinsed three times with antibody buffer (5% NDS, 0.5% Triton X-100 in PBS) for 10 min each, followed by incubation for 72 hr at 4°C with the primary antibodies in the same buffer. Primary antibodies: rabbit polyclonal anti-Tph2 (1:1000; NB100-74555; Novus Biological), rat monoclonal anti-HA (1:200; 7C9; Chromotek), goat polyclonal anti-5-HT (1:1000, ab66047; Abcam). Sections were then washed with antibody buffer three times for 10 min and incubated with secondary antibodies for 2 hr at RT. Secondary antibodies: donkey anti-rabbit IgG-Alexa Fluor 488 (1:500, ThermoFisher Scientific.), donkey anti-rat IgG-Alexa Fluor 594 (1:500, ThermoFisher Scientific), donkey anti-goat IgG-Alexa Fluor 647 (1:500, ThermoFisher Scientific). DAPI (4', 6-diamidino-2-phenylindole) was used for nuclear counterstaining.

## Image acquisition and processing

Images were collected on a Zeiss LSM 780 inverted point scanning confocal microscope with a Zeiss LD LCI Plan-Apochromat 25x/0.8 N.A. multi-immersion objective for overview images and a Zeiss Plan Apochromat 63x/1.4 N.A. oil-immersion objective for higher magnification images. Laser settings were adjusted for each sample, but kept constant throughout image collection within the same areas between *Pet1-Di* and *Control-Di* sections. The images were imported to and processed with ImageJ (Fiji distribution) for brightness and contrast adjustment, which were also kept constant between *Pet1-Di* and *Control-Di* sections.

## Experimental Set-up

Physiological measurements were obtained as described previously (*Barrett et al., 2016*; *Cummings et al., 2011a*). Briefly, ventilation was measured using a head-out plethysmograph system consisting of a body chamber and a head chamber. The body chamber (volume =~60 ml; diameter = 3 cm, length = 8.5 cm) was made from a water-jacketed glass cylinder with inlet and outlet ports that were connected to a water bath, allowing for continuous circulation of water around the chamber to maintain pup body temperature. The ambient temperature ($T_A$) of the body chamber and thus the body temperature ($T_B$) of the mouse pup were controlled by adjusting the temperature of the water circulating around the glass chamber. Both the $T_A$ and $T_B$ were continuously monitored with a thermistor probe and a fine thermocouple, respectively (Omega Engineering Inc, Stamford, CT). The head chamber (volume =~3 ml) was made from the bottom of a 50 ml plastic syringe tube (Terumo Medical, Somerset, NJ) with a piece of vinyl glove covering the larger of the two openings. A rubber gasket (Terumo Medical, Somerset, NJ) was used to hold the piece of vinyl glove in place and to secure the head chamber into the anterior end of the body chamber. A small hole was made in the center of the vinyl glove, where the snout of the mouse pup was inserted and the hole was sealed with Impregum F polyether impression material (3M, St. Paul, MN). The head chamber had an outlet port connected downstream to a pump (S-3A/I, AEI Technologies, Pittsburgh, PA) that pulled air through the head chamber at a rate of 140 ml/min. This high flow rate was chosen to prevent accumulation of $CO_2$ in the head chamber and to ensure rapid delivery of the experimental gas to the animal. The air exiting the head chamber was passed through a Nafion drying tube (PerkinElmer, Waltham, MA) before being sampled by oxygen ($O_2$) analyzers (S-3A/I, AEI Technologies, Pittsburgh, PA) in order to monitor oxygen consumption (). A pneumotach connected to the open end of the head chamber was attached to a differential pressure transducer (Validyne Engineering Corp, Northridge, CA) in order to measure respiratory activity. Experimental gases were delivered to the head chamber via the open end of a 50 ml syringe tube that was connected to the gas cylinder and then placed over the pneumotach. The pneumotach was calibrated by withdrawing and injecting 0.02 ml of air into the head chamber and the pressure signal associated with injection of this volume was integrated to determine the volume. Heart rate (HR) was monitored with a telemetric device (CTA-F40, DSI, Inc., St. Paul, MN) that consisted of 2 ECG leads that were placed on the surface of the pup's chest and held in place with a vest made from a cohesive flexible bandage (Andover, Salisbury, MA).

### Data analysis
Mortality

Mortality was calculated by assessing the total number of mice that died at any point during the assay time.

### Baseline breathing and heart rate

The LabChart application (AD Instruments Inc, Colorado Springs, CO) was used to perform data analysis on randomized, de-identified data files. Mice were continuously recorded throughout the assay. After mice were placed in the chamber, they were allowed 20 min to acclimatize. During the initial 20 min mice were exposed to room-air, with the chamber held at 35–36 ± 0.05°C (*Figure 1G*). Within the last three minutes, 30 s of stable data was taken; stability was assessed by lack of movement artifact in the breathing and heart rate traces, to obtain the baseline before silencing (see *Figure 1G* open window a). After the acclimatization period, mice were removed from the chamber briefly and injected intraperitoneally with CNO dissolved in saline (1 mg/ml) to an effective concentration of 10 mg/kg and then quickly returned to the chamber for an additional 10 min. Similar to the innate baseline measurement taken prior to CNO administration, this measurement was taken by obtaining a stable 30 s segment of data within a 3 min period immediately prior to the first asphyxic challenge (see *Figure 1H* filled window b'). The baseline breathing frequency ($f$; breaths • $\text{min}^{-1}$), tidal volume ($V_T$; ml • $\text{g}^{-1}$), minute ventilation ($\dot{V}_E$; ml • $\text{min}^{-1}$ • $\text{g}^{-1}$), oxygen consumption ($\dot{V}_{O_2}$; ml • $\text{min}^{-1}$ • $\text{g}^{-1}$), ventilatory equivalents for oxygen ($\dot{V}_E/\dot{V}_{O_2}$) and heart rate (HR) were assessed. $\dot{V}_E$ was calculated as the product of the breathing frequency $f$ and the tidal volume $V_T$. The $f$ was obtained directly from the breathing tracings, while integration of the pressure changes associated with respiratory activity was used to calculate the $V_T$. $\dot{V}_{O_2}$ was calculated as the product of the gas flow rate and the difference between the inspired and mixed expired $O_2$ normalized to body weight ([Flow (ml • $\text{min}^{-1}$)* ($F_{IO2}$ − $F_{EO2}$)] • body weight $(g)^{-1}$). HR was calculated by assessing the time between each r wave on the ECG tracings.

### Baseline Heart Rate and breathing rate variability

Using the LabChart Heart Rate Variability module, the previously defined 30 s segments of ECG and breathing traces before and during CNO exposure were analyzed. Initial peak detection used the built-in algorithm along with experimenter verification using manual selection of all r waves on the ECG trace and maximal voltage deflection on the breathing trace. Using this module, the coefficient of variation for breathing $f$ and HR was also determined. Additionally, this software was used to identify the standard deviation (SD), the major and minor axes of the Poincare plot, and the root mean square of successive deviation (RMSSD) for the interbeat and interbreath intervals.

### Heart rate and breathing rate recovery during autoresuscitation.

Apnea was induced by introducing an asphyxia-mimicking gas mixture (97% nitrogen and 3% carbon dioxide) to the 50 ml cylinder as previously described (*Barrett et al., 2016*; *Cummings et al., 2011a*; *Erickson and Sposato, 2009*). An initial examination of the time it took either genotype to develop an apnea once asphyxic conditions were introduced demonstrated no difference (23.4 ± 4.1 [standard deviation] (s) for *Pet1-Di-CNO* and 23.4 ± 4.9 [standard deviation] (s) control-CNO respectively). After the asphyxic exposure, we measured the heart rate recovery, and breathing rate recovery. Time to recover heart rate or breathing frequency to 63% of baseline ($\tau_{HR}$ and $\tau_f$, respectively) was determined to be the time between the end of asphyxia to the time it took an animal to recover and sustain for at least 3 s their heart rate or breathing frequency to 63% of their baseline heart rate or breathing frequency immediately prior to that asphyxic bout (see *Figure 4A*). The τ for each asphyxic bout was obtained using the new baseline measured immediately before each asphyxic bout (see *Figure 4A* [open window a'-d' for a-b, respectively]). Of the animals that died, some (3 of 7 *Pet1-FDi*-CNO and 1 of 1 *control*-CNO pups) nonetheless recovered heart rate to greater than 63% of the pre-bout baseline for a short period prior to death, thus data from that fatal apneic bout (referred to as the fatal bout) was included in our analyses (see *Figure 4D and E* gray-filled symbols). The other 4 *Pet1-FDi*-CNO pups that succumbed to an apneic bout never recovered to the 63% pre-bout breathing and heart rate levels during the fatal apneic bout, thus only data from bouts recovery

prior to the fatal bout were included in the graphical representations (*Figure 4B–E* black-filled symbols). One *Pet1-Di*-CNO pup, which did not die, nonetheless failed over the course of the assay to recover breathing rate to 63% of its final/4$^{th}$-bout baseline, thus we assigned as recovery time the full 331 s assay duration, albeit an underestimate. One ECG lead malfunction precluded T$_{HR}$ measurements from a single *Pet1-Di*-CNO animal for asphyxia bout 1 and 2, but which was corrected for bouts 3 and 4. For the correlation and linear regression analysis, necessarily only data from asphyxia bout recoveries with both a heart rate and breathing rate recovery were included.

## Gasp characteristics observed during the initial asphyxic bout

The first gasp was defined as the first sharp inhalation after apnea onset and during which the $O_2$ was rising determined by the $O_2$ sensor value, ~5 s, ensuring chamber $O_2$ could support autoresuscitation. Gasp size was determined as the integral of the first gasp with a voltage change greater than 0.002 V, empirically distinguishing it from smaller pressure transducer changes reflecting instead body movement artifacts. Time between gasps was calculated as the time between these 'steeple-like' voltage deflections (see *Figure 6A*). HR during gasp intervals was calculated as the mean heart rate between steeple deflections.

## Statistical analysis

The data are presented as the mean ±SD. The effects of gender and genotype on body weight (BW), $\dot{V}_E$, $\dot{V}_{O_2}$, $\dot{V}_E/\dot{V}_{O_2}$ and HR parameters at baseline before CNO injection (*Figure 1G*, open window *a*) were assessed by a two-tailed Student t-test comparisons. To assess the effects of CNO on the experimental and control groups, paired two-tailed Student t-tests were performed on the baseline homeostatic characteristics (HR, $\dot{V}_E$, $\dot{V}_{O_2}$, $\dot{V}_E/\dot{V}_{O_2}$, and heart rate and breathing rate variability characteristics) of each group before vs. during CNO exposure (*Figure 1G* open window *a* and 1 H bar *b'*, respectively). To test the hypothesis of increased mortality in *Pet1-FDi-CNO* pups, we applied a one-tailed Fisher Exact test with Lancaster's mid-p correction (*Biddle and Morris, 2011*) due to the previous assumption that *Pet1* raphe neuron disruption would increase mortality (*Barrett et al., 2016*). The odds ratio for pup death as an outcome of asphyxic apnea in the face of *Pet1-Di*-CNO versus control-*Di*-CNO was calculated as follows: (A/C)/(B/D), where A = *Pet1 Di*-CNO pups that died, B = *Pet1 Di*-CNO pups that survived, C = Control *Di* pups that died, and D = Control *Di* pups that survived. OpenEpi version three was used to perform the one-tailed Fisher Exact test (*Sullivan et al., 2009*). To analyze breathing and heart rate recoveries across asphyxia bouts, we used a two-way analysis of variance (ANOVA) with asphyxia bout and genotype as variables, with Tukey's test for multiple comparison correction *post hoc* and unpaired t-tests. A linear regression model was run to assess the relationship between BR versus HR recovery. Additionally, a Runs test was used to assess whether these data had a nonrandom linear relationship. To compare first-gasp responses, a one-way ANOVA test was applied. To assess the relationship between gasp interval duration and genotype survival between gasp intervals, we applied a repeated measures two-way ANOVA with gasp interval and genotype/survival as variables; *post hoc* analyses employed Tukey's test for multiple comparisons to determine the effect of asphyxia on the different groups. We used the same analysis to assess the relationship between HR during gasp interval and genotype survival. To test for independent effects of homeostatic characteristics on mortality, a logistic regression model was fit with mortality (yes/no) as the outcome and homeostatic characteristics as predictors, controlling for genotype. Variance of $\dot{V}_E/\dot{V}_{O_2}$ measurement was analyzed using an F test. All graphs and all other statistical analyses were performed using GraphPad Prism version 7.0 c for Mac OS X, GraphPad Software, La Jolla California USA, www.graphpad.com.

## Acknowledgements

Funding for this study was provided by National Institute of Health (NIH) Program Project Grant HD036379 (Program Director: SM Dymecki; Project PIs: EE Nattie and SM Dymecki) and the Howard Hughes Medical Institute Gilliam Fellowship, National Institute of General Medical Sciences award number T32GM07753, and Society for Neuroscience, Neuroscience Scholars Program (RT Dosumu-Johnson). The content is solely the responsibility of the authors and does not necessarily represent the official views of the National Institute of Child Health and Human Development, National

Institute of General Medical Sciences, or the National Institutes of Health more generally. The authors gratefully acknowledge the statistical assistance of Felicia L Trachtenberg, Ph.D., New England Research Institutes, Watertown, Massachusetts, as well as the technical assistance of Jia Jia Mai at Harvard Medical School (HMS), the microscopy services of MicRoN (Microscopy Resources on the North Quad, Dr. Paula Montero Llopis and Ryan Stephansky) at HMS, and Dr. Olga V. Alekseynko of HMS for independent randomization prior to analysis.

## Additional information

### Funding

| Funder | Grant reference number | Author |
|---|---|---|
| Howard Hughes Medical Institute | Gilliam Fellowship | Ryan T Dosumu-Johnson |
| National Institutes of Health | HD036379 | Ryan T Dosumu-Johnson<br>Andrea E Cocoran<br>YoonJeung Chang<br>Eugene Nattie<br>Susan M Dymecki |
| National Institute of General Medical Sciences | T32GM07753 | Ryan T Dosumu-Johnson |
| Society for Neuroscience | Neuroscience Scholars Fellowship | Ryan T Dosumu-Johnson |

The funders had no role in study design, data collection and interpretation, or the decision to submit the work for publication.

### Author contributions

Ryan T Dosumu-Johnson, Conceptualization, Data curation, Formal analysis, Validation, Investigation, Visualization, Methodology, Writing—original draft, Writing—review and editing; Andrea E Cocoran, Conceptualization, Investigation, Methodology, Writing—review and editing; YoonJeung Chang, Validation, Investigation, Methodology, Writing—review and editing; Eugene Nattie, Conceptualization, Formal analysis, Supervision, Funding acquisition, Writing—review and editing; Susan M Dymecki, Conceptualization, Resources, Data curation, Formal analysis, Supervision, Funding acquisition, Validation, Visualization, Methodology, Writing—original draft, Project administration, Writing—review and editing

### Author ORCIDs

Ryan T Dosumu-Johnson https://orcid.org/0000-0002-0120-9565
Susan M Dymecki http://orcid.org/0000-0003-0910-9881

### Ethics

Animal experimentation: All experimental protocols were approved at Harvard Medical School (HMS) and the Geisel School of Medicine at Dartmouth by the respective Institutional Animal Care and Use Committees (IS00000231-3 and 2035, respectively) and the HMS Committee on Microbiological Safety (15-225), and were in accordance with the animal care guidelines of the National Institutes of Health.

### Decision letter and Author response

Decision letter https://doi.org/10.7554/eLife.37857.021
Author response https://doi.org/10.7554/eLife.37857.022

## Additional files

### Supplementary files
• Transparent reporting form
DOI: https://doi.org/10.7554/eLife.37857.019

### Data availability
Data generated or analysed during this study are included in the manuscript.

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
