## [Decision Letter]

Thank you for submitting your article "Acute perturbation of *Pet1-*neuron activity in neonatal mice impairs cardiorespiratory homeostatic recovery" for consideration by *eLife*. Your article has been reviewed by three peer reviewers, including Jan-Marino Ramirez as the Reviewing Editor and Reviewer #1, and the evaluation has been overseen by Gary Westbrook as the Senior Editor. The following individuals involved in review of your submission have agreed to reveal their identity: Matthew Hodges (Reviewer #2); Christopher Wilson (Reviewer #3).

The reviewers have discussed the reviews with one another and the Reviewing Editor has drafted this decision to help you prepare a revised submission.

Summary:

The study by Drs. Dosumu-Johnson and co-workers constitutes an important step towards a better understanding of the neuronal mechanisms underlying autoresuscitation, as well as the mechanisms that lead to a failure to autoresuscitate. These are exceedingly important topics, because SIDS has been associated with failure to arouse and autoresuscitate in response to a hypoxic and hypercapnic challenge. Moreover, there is ample evidence that disturbances in serotonergic mechanisms play critical roles in the events leading to the sudden death. Although, it is well established that serotonin is an important neuromodulator for breathing, in particular in the so-called pre-Bötzinger complex, we don't understand how disturbances in 5-HT lead to failed autoresuscitation.

This study provides not only important clues into the mechanisms leading to failed respiratory response, but also to disturbed cardiorespiratory coupling, which is ultimately what seems to be responsible for the death. The authors use cutting-edge genetic tools to dissect the neuronal mechanisms underlying cardiorespiratory coupling and the homeostatic regulation of breathing. Interestingly, the authors observe a cardiorespiratory uncoupling because acute manipulation of *Pet1*^+^ neurons affects primarily the respiratory response while keeping intact the heart rate recovery. Thus, the study provides strong support that these two components (respiration/heart control) are differently controlled during the autoresuscitation process. This is a fascinating (perhaps even unexpected), yet well documented finding.

Essential revisions:

1) As expected the authors show a dramatic decrease in minute ventilation in response to CNO in the double transgenic *Pet1-Di* pups. Interestingly, the mice with the highest frequency showed the most dramatic inhibitory effect – which is very interesting. The authors suggest that the serotonin neurons function as homeostatic regulators that become most relevant when the frequency deviates most (Table 2 and Figure 6). The authors discuss alternative explanations, and to prove this hypothesis would require experiments that are out of the scope of this already elegant study. However, the authors should be more cautious in the Discussion, e.g.: "Evidence provided here suggests…".

2) The Figure 3B has several "spikes" in HR changes – we assume these are artifacts. Perhaps it would be good to remove obvious artifacts (if this is possible) to better see the changes in cardiorespiratory coupling.

3) The authors refer to "anoxia" throughout the text, but their gas mixture created not only hypoxic conditions but also altered CO_2_. Why was the specific gas mixture (97% N_2_, 3% CO_2_) used as the anoxic challenge? Is it simply that this is a standard anoxic mixture used to induce asphyxic apnea and subsequent gasping? It is curious that the mixture contained 3% CO_2_, which would best mimic asphyxia and not anoxia, but further explanation should be provided in the methods to elaborate on this experimental choice of exposure.

4) Figure 5 is a very elegant way to describe the differences in the breathing/HR response. Would it be possible to add an original trace to qualitatively show this differential breathing/HR response, given that this is a main message of the paper? They have done this for Figure 6, and could modify their Figure 5 similarly by adding trace examples.

5) A major conclusion from the data are that some *Pet1-Di* pups, when given CNO to inhibit their (largely) serotonergic neuronal activity, decrease ventilation at rest while others show a modest stimulation of ventilation. This can be interpreted as *Pet1* neurons allowing for a greater dynamic range around a homeostatic set point. While a valid conclusion, another explanation relates to the metabolic rate. If for example the V˙E/V˙O2 ratio was uniformly decreased, while V˙E was not uniformly affected, this may change the interpretation of the overall effect of 5-HT neuronal inhibition on ventilation. We encourage the authors to analyze the effects of CNO on the V˙E/V˙O2 ratio in control-*Di* and *Pet1-Di* mice to confirm that indeed there are a mix of inhibitory and excitatory effects on the integration of ventilation and metabolic rate. Please consider looking at the effects of CNO on V˙O2 in the same way as V˙E and HR or even the coefficient of variation, as it would further strengthen the authors' conclusions regarding dynamic range and the influence of *Pet1* neurons. This additional analysis will likely further strengthen the conclusion that silencing of 5-HT neurons has greater effects on the respiratory vs. cardiovascular systems.

6) Another important consideration is that CNO injection alone, in the absence of anoxia, did not induce apnea in *Pet1-Di* pups. This is somewhat surprising given the evidence that *Pet1* knockout (Erickson et al., 2007) and *lmx1b* conditional knockout mice (Hodges et al., 2009) develop spontaneous apnea and have increased mortality during development. In addition, increasing concentrations of 5-HT or neurokinin-1 receptor antagonists suppress or eliminate respiratory rhythm in situ (Hodges et al., JON, 2009). Was ventilation or HR destabilized in *Pet1-Di* mice when given CNO? If you just give CNO and measure ventilation and HR over time without anoxia, do they develop additional phenotypic changes? Also relevant to this concept are the data (albeit rare) of prospective traces from human SIDS cases, which show spontaneous apnea and bradycardic events preceding a final event. Does CNO induce increases (or decreases) in breathing or HR variability? SIDS has been associated with decreased HR variability, could the authors discuss their findings in this context?

7) The results indicate that *Pet1* neurons are critical for proper cardiorespiratory coupling, particularly in the context of mounting an autoresuscitatory response to anoxia and ultimately survival. Perhaps the authors could discuss possible brainstem regions that may be responsible for this outcome. Is it at the level of the pre-Bötzinger complex and the interactions with the *N. ambiguus* (cardiovagal neurons), or the nucleus of the solitary tract, an important site of integration of peripheral O_2_ chemoreflex? It seems that this could be a major point discussed at the end of the manuscript that is highly relevant to the data presented, especially in the context of cardiorespiratory uncoupling in this elegant model.

8) Was there a difference in the total number of neurons expressing *Pet1-Di* in the animals that showed a difference in V˙E? Figure 1 shows labeled cells in the DRN, RMg, and raphe but we did not see stereologically verified counts of the number of cells expressing DiHA. It may be that the animals hyperventilating prior to CNO are already deficient in their control circuitry. Since the authors have the labeled images, counts of the labeled cells in this cohort of animals should be straightforward.

9) The authors did not measure the breath-to-breath variability. Adding such a characterization would provide further insights into the reported data. Even something as simple as coefficient of variation for each epoch measured would be helpful for understanding the perturbations leading to autoresuscitative failure. Measures of heart rate variability (in addition to reported changes in HR) would be of interest as well since the ECG data is available.

10) How many pups in each litter died before P8? We could not find this information in the text. Yet others (Erickson et al., 2007, and later papers) have reported higher mortality in the first few days of life for *Pet1* KO animals. Providing some "baseline" idea of mortality before P8 would be informative.

---

## [Author Response]

Essential revisions:1) As expected the authors show a dramatic decrease in minute ventilation in response to CNO in the double transgenic Pet1-Di pups. Interestingly, the mice with the highest frequency showed the most dramatic inhibitory effect – which is very interesting. The authors suggest that the serotonin neurons function as homeostatic regulators that become most relevant when the frequency deviates most (Table 2 and Figure 6). The authors discuss alternative explanations, and to prove this hypothesis would require experiments that are out of the scope of this already elegant study. However, the authors should be more cautious in the Discussion, e.g.: "Evidence provided here suggests…".

Discussion text has been duly revised, including but not limited to what is now the first paragraph of the subsection “Pet1 neurons shape the neonatal P8 cardiorespiratory homeostatic set point and dynamic range”.

2) The Figure 3B has several "spikes" in HR changes – we assume these are artifacts. Perhaps it would be good to remove obvious artifacts (if this is possible) to better see the changes in cardiorespiratory coupling.

Minimization of tracing artifact was achieved by altering the bin size.

3) The authors refer to "anoxia" throughout the text, but their gas mixture created not only hypoxic conditions but also altered CO_2_. Why was the specific gas mixture (97% N_2_, 3% CO_2_) used as the anoxic challenge? Is it simply that this is a standard anoxic mixture used to induce asphyxic apnea and subsequent gasping? It is curious that the mixture contained 3% CO_2_, which would best mimic asphyxia and not anoxia, but further explanation should be provided in the methods to elaborate on this experimental choice of exposure.

Indeed, the intent of the specific gas mixture is to induce asphyxic apnea (hypoxia and hypercapnia combined) – conditions often associated with SIDS. We have revised the terminology throughout, included a more detailed explanation around choice of exposure, and continue to include relevant citations.

4) Figure 5 is a very elegant way to describe the differences in the breathing/HR response. Would it be possible to add an original trace to qualitatively show this differential breathing/HR response, given that this is a main message of the paper? They have done this for Figure 6, and could modify their Figure 5 similarly by adding trace examples.

We have added example original traces to qualitatively illustrate the differential breathing/HR response seen in the *Pet1-Di*-CNO pups. These traces form Figure 5—figure supplement 1; the associated, revised primary Figure 5 lacks room for these traces as it is now comprised of 15 panels to fulfill reviewers’ request to include scatter plots for all apnea bouts and each genotype.

*5) A major conclusion from the data are that some Pet1-Di pups, when given CNO to inhibit their (largely) serotonergic neuronal activity, decrease ventilation at rest while others show a modest stimulation of ventilation. This can be interpreted as Pet1 neurons allowing for a greater dynamic range around a homeostatic set point. While a valid conclusion, another explanation relates to the metabolic rate. If for example the*
V˙E/V˙O2
*ratio was uniformly decreased, while*
V˙E
*was not uniformly affected, this may change the interpretation of the overall effect of 5-HT neuronal inhibition on ventilation. We encourage the authors to analyze the effects of CNO on the*
V˙E/V˙O2
*ratio in control-Di and Pet1-Di mice to confirm that indeed there are a mix of inhibitory and excitatory effects on the integration of ventilation and metabolic rate. Please consider looking at the effects of CNO on*
V˙O2V˙O2
*in the same way as*
V˙E
*and HR or even the coefficient of variation, as it would further strengthen the authors' conclusions regarding dynamic range and the influence of Pet1 neurons. This additional analysis will likely further strengthen the conclusion that silencing of 5-HT neurons has greater effects on the respiratory vs. cardiovascular systems.*

Added as Figure 2—figure supplement 3 are analyses of V˙E/V˙O2 and of V˙O2 in *control-Di-* and *Pet1-Di* mice in relation to CNO exposure. For V˙E/V˙O2 data, we see no overall statistically significant differences pre- versus during CNO exposure for either genotype (Figure 2—figure supplement 3A, A’); there is, though, a trend for animals with a lower V˙E/V˙O2pre-CNO to manifest an increase in V˙E/V˙O2during the CNO exposure, and vice versa, those with a high pre-CNO value show a trend to decrease during the CNO exposure, albeit not in all cases. These findings, albeit modest, are in line with the model that Pet1 neurons allow for a greater dynamic range around a homeostatic set point.

Baseline V˙O2 measurements were less straightforward, showing in both genotypes a subtle decrease following CNO and return to the plethysmograph chamber. We speculate that this non-specific V˙O2 effect reflects relaxation in and habituation to the chamber at this advancing time point in the assay resulting in a subtle lowering of metabolic rate. It could also reflect the very real challenge in accurately measuring V˙O2 (as compared to the other cardiorespiratory parameters) for such tiny mouse pups and/or reflect a modest effect of CNO itself (independent of *Di* expression) on P8 pup metabolic rate. Despite these caveats, pups demonstrating the largest change in V˙O2 pre- versus during CNO were also the animals that showed the highest pre-CNO baseline V˙O2(Figure 2—figure supplement 3D, D’); this is reminiscent of the V˙E data, which could be interpreted to add further support to the dynamic range model proposed for a role of *Pet1* neurons in neonates.

6) Another important consideration is that CNO injection alone, in the absence of anoxia, did not induce apnea in Pet1-Di pups. This is somewhat surprising given the evidence that Pet1 knockout (Erickson et al., 2007) and lmx1b conditional knockout mice (Hodges et al., 2009) develop spontaneous apnea and have increased mortality during development. In addition, increasing concentrations of 5-HT or neurokinin-1 receptor antagonists suppress or eliminate respiratory rhythm in situ (Hodges et al., JON, 2009). Was ventilation or HR destabilized in Pet1-Di mice when given CNO?

Standard deviation of interbreath interval was significantly decreased in *Pet1-Di* animals upon CNO administration; CNO had no discernable effect on this parameter in Control-Di animals. Thus the stability of the respiratory rate was altered upon perturbation of *Pet1* neurons. No alterations were observed in standard deviation of HR interbeat interval in either genotype. Nor could we uncover differences in the coefficient of variation in respiratory rate or heart rate upon CNO/Di-mediated perturbation of *Pet1* neurons.

If you just give CNO and measure ventilation and HR over time without anoxia, do they develop additional phenotypic changes?

We look forward to addressing this in work that falls outside the scope of the present experimental design, which was to assess effects on the autoresuscitation response. Longer-duration ventilation and HR traces would allow for proper assessment.

Also relevant to this concept are the data (albeit rare) of prospective traces from human SIDS cases, which show spontaneous apnea and bradycardic events preceding a final event. Does CNO induce increases (or decreases) in breathing or HR variability? SIDS has been associated with decreased HR variability, could the authors discuss their findings in this context?

The additional analyses show only an effect on SD of interbreath interval, suggesting that inhibition of *Pet1* neurons decreases this deviation, i.e. decreases breathing variability, perhaps reflecting a decrease in scope of breathing range and thus a diminished capacity to respond to respiratory stressors. No significant findings around HR variability were observed.

Note, experiments were designed to examine recovery from induced apneas, with little opportunity to detect spontaneous apneas.

7) The results indicate that Pet1 neurons are critical for proper cardiorespiratory coupling, particularly in the context of mounting an autoresuscitatory response to anoxia and ultimately survival. Perhaps the authors could discuss possible brainstem regions that may be responsible for this outcome. Is it at the level of the pre-Bötzinger complex and the interactions with the N. ambiguus (cardiovagal neurons), or the nucleus of the solitary tract, an important site of integration of peripheral O_2_ chemoreflex? It seems that this could be a major point discussed at the end of the manuscript that is highly relevant to the data presented, especially in the context of cardiorespiratory uncoupling in this elegant model.

We agree about the importance of adding text that directs thinking down the path of downstream responsible brain regions modulated by *Pet1* neurons. Thus we have added text in the Discussion, subsection “Initial response characteristics correlate with future apnea-related mortality”, last paragraph; albeit given limitations in space, it is just a hint to compel curiosity and discourse beyond the scope of this manuscript.

8) Was there a difference in the total number of neurons expressing Pet1-Di in the animals that showed a difference in V_E_? Figure 1 shows labeled cells in the DRN, RMg, and raphe but we did not see stereologically verified counts of the number of cells expressing DiHA. It may be that the animals hyperventilating prior to CNO are already deficient in their control circuitry. Since the authors have the labeled images, counts of the labeled cells in this cohort of animals should be straightforward.

To address this point, we have added Figure 1—figure supplement 1, which presents additional examples of DiHA staining in the medulla of *Pet1-Di* pups. Qualitatively, we discern no difference in marked cells (number and distribution) across independent litter-derived *Pet1-Di* pups. Of course, medullary immunodetection of DIHA is not quantitative, reflects only soma localization while much is likely localized to axon terminals given what is known generally about preferential shuttling of muscarinic receptors, and soma DiHA detection may not directly correlate with the severity of neuronal inhibition.

We do not have the precise brain tissue from the experimental cohorts subjected to the asphyxia bouts; thus, we cannot relate stereologically verified counts of cells to the severity of cardiorespiratory phenotype.

9) The authors did not measure the breath-to-breath variability. Adding such a characterization would provide further insights into the reported data. Even something as simple as coefficient of variation for each epoch measured would be helpful for understanding the perturbations leading to autoresuscitative failure. Measures of heart rate variability (in addition to reported changes in HR) would be of interest as well since the ECG data is available.

In response to this point, we have added Figure 2—figure supplement 1, Figure 6—figure supplement 1, and Figure 6—figure supplement 2 in which the following parameters were analyzed pre- versus during CNO exposure for Control-*Di* pups and *Pet1-Di* pups: coefficient of variation in respiratory rate, coefficient of variation in heart rate (Figure 2—figure supplement 1); standard deviation (SD) of interbeat interval for HR, SDxSD of Poincare major and minor axis for HR interbeat interval, root mean square of SD for HR interbeat interval (Figure 6—figure supplement 2); SD of interbreath interval for respiratory rate, SDxSD of Poincare major and minor axis for respiratory interbreath interval, root mean square of SD for interbreath interval (Figure 6—figure supplement 1). The only statistically significant difference discerned was in the SD of interbreath interval for respiratory rate, suggesting a narrowing of rate variability upon perturbation of *Pet1* neurons. Perhaps this manifests physiologically as a vulnerability to respiratory stress. Indeed in broad terms, lesser variability of respiratory rate or HR can be seen as a less healthy control system. E.g., in studies of HR variability in ICU patients, lower variability predicts poor outcome. Perhaps, affected 5-HT neurons are part of a complex control system whose overall ‘health’ can be measured by variability analysis.

10) How many pups in each litter died before P8? We could not find this information in the text. Yet others (Erickson et al., 2007, and later papers) have reported higher mortality in the first few days of life for Pet1 KO animals. Providing some "baseline" idea of mortality before P8 would be informative.

Baseline mortality before P8 was zero. This was expected given no intended perturbation was induced prior to P8, rather all pups (regardless of genotype) were effectively wild type in the absence of CNO administration. By contrast, the mentioned constitutive knock-out animals of prior studies (e.g. *Pet1^-/-^* animals, Erickson et al., 2007) were severely defective from mid-gestation onward, including birth through P8 and of course into adulthood.